# Evolution and subfunctionalization of *CIPK6* homologous genes in regulating cotton drought resistance

Weinan Sun[1,4], Linjie Xia[1,4], Jinwu Deng[1], Simin Sun[1], Dandan Yue[1], Jiaqi You[1], Maojun Wang [1,2], Shuangxia Jin [1,2], Longfu Zhu [1,2], Keith Lindsey [3], Xianlong Zhang [1,2] & Xiyan Yang [1,2] ✉

The occurrence of whole-genome duplication or polyploidy may promote plant adaptability to harsh environments. Here, we clarify the evolutionary relationship of eight *GhCIPK6* homologous genes in upland cotton (*Gossypium hirsutum*). Gene expression and interaction analyses indicate that *GhCIPK6* homologous genes show significant functional changes after polyploidy. Among these, *GhCIPK6D1* and *GhCIPK6D3* are significantly up-regulated by drought stress. Functional studies reveal that high *GhCIPK6D1* expression promotes cotton drought sensitivity, while *GhCIPK6D3* expression promotes drought tolerance, indicating clear functional differentiation. Genetic and biochemical analyses confirm the synergistic negative and positive regulation of cotton drought resistance through GhCBL1A1-GhCIPK6D1 and GhCBL2A1-GhCIPK6D3, respectively, to regulate stomatal movement by controlling the directional flow of K+ in guard cells. These results reveal differentiated roles of *GhCIPK6* homologous genes in response to drought stress in upland cotton following polyploidy. The work provides a different perspective for exploring the functionalization and subfunctionalization of duplicated genes in response to polyploidization.

Whole-genome duplication (WGD, polyploidization), which directly doubles or trebles the entire genome, is the main form and mechanism of gene duplication[1–3], and is found in the ancestors of many species-rich groups, such as the *Compositae*, *Cruciferae*, *Gramineae* and *Leguminosae*[4–6]. Studies have shown that the time of occurrence of WGDs in various stages of angiosperm evolution is not random, indicating that WGDs may play a role in environmental adaptation[7]. The replicated genes are considered to be an important force in species formation, adaption and diversity during plant evolution[8,9].

Polyploidy replicates tens of thousands of genes simultaneously by adding one or more extra genomes, providing not only a large amount of primitive genetic material for plant evolution but also opportunities for plant evolution through species variation and diversification[7,10]. A large number of duplicated genes are produced during polyploidization, acting as a very important driver in the evolution of gene families, leading to changes in gene regulatory networks (GRNS), and allowing functional differentiation and innovation, which may potentially contribute to plant adaptation[10,11]. Therefore, polyploidy is considered to be the driving force of evolution and diversity in plants, as well as a key factor in the domestication of crops such as wheat, rape, soybeans and cotton[12–14]. Copied genes are either lost or retained through selection, while retained genes form the basis of gene innovation[15–18]. Importantly, numerous studies have shown that the protein kinase family is one of the most typical and abundant types of replication-retained genes[19–21].

[1]National Key Laboratory of Crop Genetic Improvement, Huazhong Agricultural University, Wuhan, Hubei 430070, P. R. China. [2]Hubei Hongshan Laboratory, Wuhan, China. [3]Department of Biosciences, Durham University, Durham, UK. [4]These authors contributed equally: Weinan Sun, Linjie Xia. ✉e-mail: yxy@mail.hzau.edu.cn

Calcineurin B-like protein (CBL) and CBL-interacting protein kinase (CIPK) form the CBL-CIPK signaling network, which plays an important role in abiotic stresses of plants, including salt, drought and cold[22–24]. One core conserved region in CBL proteins is four EF-hands, responsible for sensing calcium signals; while the conserved PFPF/FPSF motif, located at the C-terminus, plays a key role in promoting CIPK phosphorylation through direct protein-protein interaction[25,26]. CIPKs contain a NAF/FISL motif at the C-terminus, which is mainly responsible for binding to CBLs and relieving the self-inhibitory kinase activity[27]. Evolutionary analyses of *CBLs* and *CIPKs* in Arabidopsis and rice showed that the individual branches of two gene families diverged early, and even the recently replicated family members may be performing different functions[11,28]. Gene expression analysis showed that relatively overrepresented replicated genes tend to exhibit asymmetric expression, thus avoiding competition[21].

Systematic genomic evidence has shown that the conservation of NAF motifs promotes the formation of CBL-CIPK modules and helps plants evolve, replicate and amplify from algal ancestors through enhanced adaptation to abiotic stress. They evolve from a single CBL-CIPK complex in algae to complex and diverse CBL-CIPK signal networks as the core component of calcium signal transduction, and CIPKs are widely involved in the response of plants to drought stress[23,29,30]. Overexpression of *BdCIPK31* in tobacco reduced water loss under dehydration and improved drought resistance[31]. Heterologous expression of *TaCIPK2*, *TaCIPK23*, *TaCIPK27* in wheat could enhance drought resistance in an ABA-dependent pathway[32,33]. In apples, overexpression of *MdCIPK22* improved the accumulation of sugars in vacuoles to improve drought resistance[34]. The root phenotype analysis of more than 300 maize inbred lines under different water conditions indicated that *ZmCIPK3* might enhance drought resistance by regulating root elongation[35]. The synergistic effect of *CIPK* and *CBL* represents a key module to regulate the response of plants to drought and other stresses. The loss of CBL2/3-CIPK9/17 system function in Arabidopsis guard cells enhances drought resistance[36]. The interaction of OsCBL8-OsCIPK17 in rice not only regulates the growth of seedlings but also improves multiple stress resistance of rice, including drought resistance and disease resistance[37]. CBL2-CIPK6 interaction has also been found in cotton, and regulates the homeostasis of sugars by regulating the sugar transporter TST2 located in the vacuole membrane, thereby enhancing stress resistance[38].

Comparative analysis has shown that allopolyploid cotton (AD genome) shows a greater flexibility in response to mild salt stress, and innovation in phenotypic response to salt stress, compared with its diploid ancestors (A genome and D genome)[39]. A study of the genome-wide transcriptome of allopolyploid plants found that the expression of homologous genes is biased and divergent, which may lead to the differentiation of biological functions[40–43]. Through the analysis of 376 upland cotton genomes linked to the transcriptome data of 2215 populations at six-time points of fiber development, it was found that cotton subgenomic homologous genes showed dynamic and partial expression at different stages of fiber development, providing a theoretical basis for the improvement of cotton fiber quality[44]. In allohexaploid wheat, the differential response of homologous gene expression to stress environments not only affects the accumulation of chlorophyll but also infection by *Fusarium oxysporum*[43,45]. Therefore, the evolution of plant heteropolyploidy is not only closely related to stress response but also accompanied by evolutionary functional innovation[46].

In this work, we track the evolution and functional differentiation of eight homologous *CIPK6* genes in upland cotton. First, we analyze the phylogenetics of eight *CIPK6* genes in 25 representative species, including the *Gossypium* genus, to determine the evolutionary relationship between retention and replication. We then study expression and interaction of *GhCIPK6s* in different *Gossypium* species, and find

that *GhCIPK6D1* and *GhCIPK6D3* have different expression and interaction patterns with *CBL* family genes. Finally, we find that the two genes regulate drought resistance differentially, and explore the underlying mechanism of this differential regulation.

## Results

### Evolution of *CIPK6* gene family in *Gossypium*
In our previous study, eight members of the *GhCIPK6* homologous genes were systematically identified and named in upland cotton[26]. In order to trace the evolution of the *CIPK6* homologous genes in *Gossypium*, we carried out evolutionary tree analysis by selecting 23 representative species that have experienced WGD, together with two diploid parents of cotton (*G. arboretum* and *G. raimondii*). *CIPK6* genes appeared after ferns and before the divergence of monocotyledons and dicotyledons. After two rounds of WGD, four *CIPK6* genes were eventually retained in each of *G. arboretum* and *G. raimondii* (Fig. 1a). We then conducted phylogenetic analysis on the sequence of evolutionary replication events for eight *GhCIPK6* genes in cotton. The results show that *CIPK6A1/D1* first appeared in diploid species, and then replicated to form *CIPK6A4/D4*, then *CIPK6A2/D2* and *CIPK6A3/D3* were copied, and eight *GhCIPK6* genes were formed in *G. hirsutum* following hybridization of diploid species (Fig. 1b, Supplementary Fig. 1a). This evolutionary pattern was also verified using other four tetraploid cotton species (*G. barbadense*, *G. tomentosum*, *G. mustelinum* and *G. darwinii*) (Supplementary Fig. 1b).

The collinearity, sequence similarity and evolutionary rate (Ka/Ks) of *CIPK6* genes in *Gossypium* were also analyzed. It showed that *GhCIPK6s* correspond one-to-one with *GaCIPK6s* and *GrCIPK6s* on chromosome positions, and there was no exchange, gene rearrangement or translocation from diploid species in *G. hirsutum* (Supplementary Fig. 2a). We further analyzed gene duplication and evolution in polyploidization-diploidization cycles in plants. The results indicate that the main mechanism for replication is WGD, rather than tandem duplication (TD), proximal duplication (PD), transposed duplication (TRD), or dispersed duplication (DSD) in *G. arboretum* and *G. raimonddi* (Supplementary Fig. 2b). Interestingly, by analyzing the Ka/Ks of *CIPK6* genes in *Gossypium*, it is found that the evolution of *GhCIPK6D1* from diploid to upland cotton was influenced by natural selection, which may result in functionalization in response to environmental stress (Supplementary Fig. 3).

### Expression patterns of *Gossypium CIPK6* genes under drought stress
Large-scale gene replication often leads to changes in gene regulatory networks. We therefore analyzed the expression patterns of *CIPK6* genes between upland cotton and two diploid species under normal watering and induced soil drought conditions. Under normal conditions, the expression of *GhCIPK6* genes showed significant changes compared with the *GaCIPK6* and *GrCIPK6* genes (Fig. 1c–f). Importantly, under drought stress, the drought-induced expression patterns of some *CIPK6* genes changed from diploid cotton to tetraploid cotton. In the At subgenome, the drought expression patterns of *6A2* and *6A1* showed opposite trends, whereby the *6A1* expression pattern remained up-regulated and *6A2* remained down-regulated (Fig. 1c and e), but *6A3* was the same as *6A4* in the tetraploid, while both *6A4* and *6A3* changed from down-regulated to no expression change under drought (Fig. 1d and f). In the Dt subgenome, the expression of *6D1/6D2/6D3* remained up-regulated by drought, and the expression of *6D4* changed from up-regulated to down-regulated (Fig. 1c–f). It was found that *GhCIPK6D1* and *GhCIPK6D3* were significantly induced by drought stress, with *GhCIPK6D3* showing a relative high level even under normal watered conditions, while *GhCIPK6D1* showed a very low expression under watering, but sharply increasing a high level under drought stress. Whether the biological functions of these two genes are identical need further investigation.

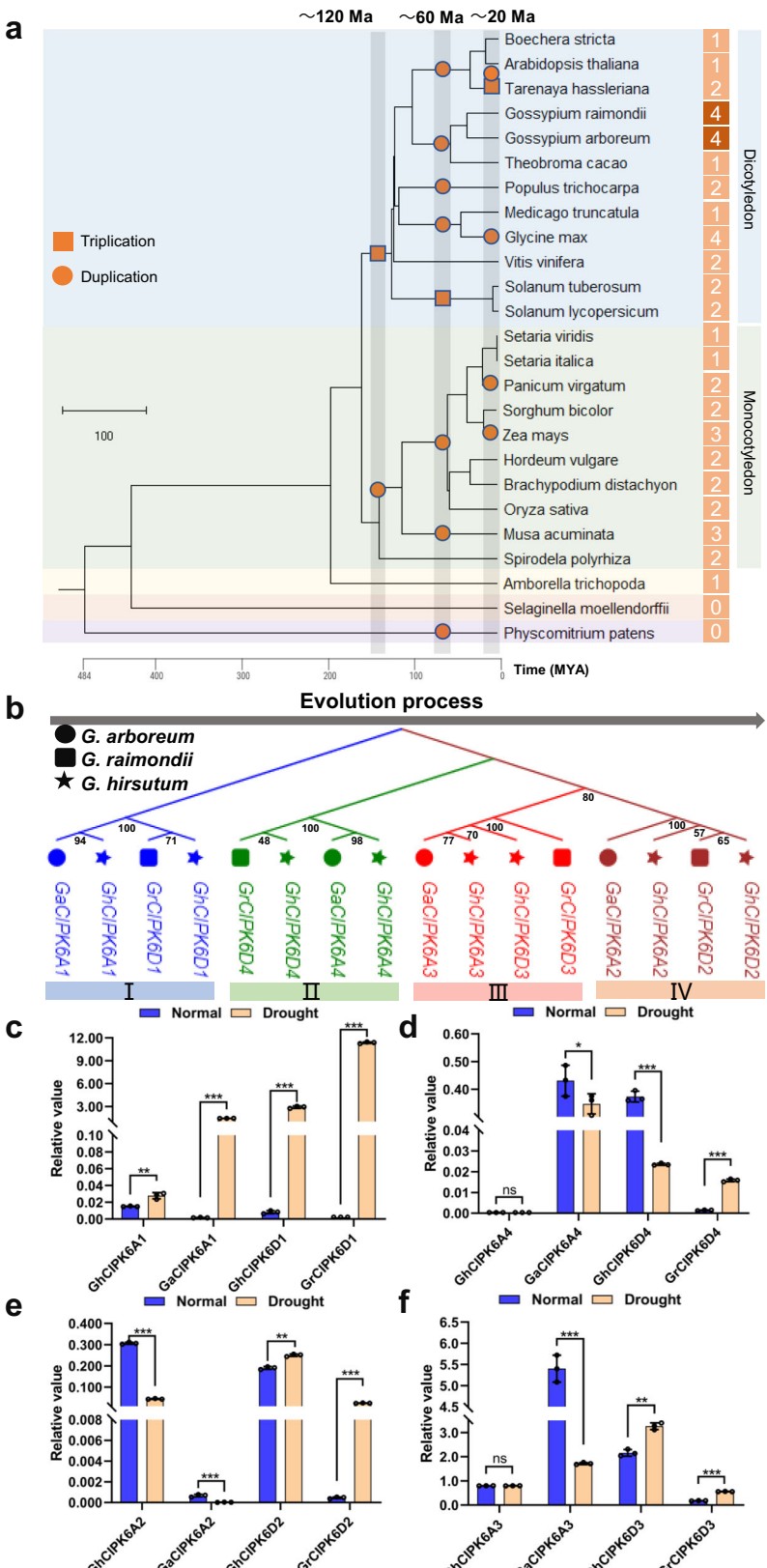

In order to explore the reasons for the differences in expression among members of the *CIPK6* homologous genes, we selected their promoter sequences (2000 bp) to analyze the cis-acting elements. The results show that the type, number and location of all elements on the promoter of each member of *CIPK6s* gene were different (Supplementary Fig. 4), which may be the basis of the difference of

expression among *CIPK6* homologous genes. At the same time, the difference in drought stress-related elements in the *CIPK6* gene promoters were investigated. It was found that *GhCIPK6D1* contains continuous drought-induced MYB binding sites, and *GhCIPK6D3* contains consecutive elements involved in the ABA response compared to *GrCIPK6D1* and *GrCIPK6D3* (Supplementary

**Fig. 1 | Evolution and expression pattern analyses of *CIPK6* genes in *Gossypium*. a** Identification and evolution of *CIPK6s* in *G. raimondii* and *G. aboreum* after WGDs through evolutionary history. The species tree was constructed based on TimeTree website (http://timetree.org/) and save as 'nwk' format (a text format containing node and branch information of a tree), and we used MCScanX to draw and embellish the species tree. The 25 species (including two diploid cotton) were obtained from their genome website. Three periods (-120, -66, and <20 Ma) with WGDs are indicated in gray. The orange square represents triplication, and circle represents duplication. The number at the end of the tree represents the number of *CIPK6* in the species. **b** Inferring of evolution order of eight *GhCIPK6* genes in *G. hirsutum* after tetraploidization. The phylogenetic tree was completed with MAGA 7. Circle represents *G. aboreum*, square represents *G. raimondii*, and star represents *G. hirsutum*. Different color branches represent different *CIPK6* subfamily gene members. The gray arrow and capital numerals indicate an inference about the evolution order. The different numbers at the node of the evolutionary tree are the bootstrap value. They are used to assess the reliability of evolutionary tree branches. **c–f** Analysis of expression of each pair of homologous genes in upland cotton and their parent cotton under normal and drought stress, 6A1/6D1 (**c**), 6A4/6D4 (**d**), 6A2/6D2 (**e**), 6A3/6D3 (**f**). The blue column represents normal watered conditions, and orange column represents drought treatment. In (**c–f**), data are means ± SD ($n = 3$ biological replicates). Significant difference analysis used two-tailed Student's $t$-test (*$P < 0.5$, **$P < 0.01$, ***$P < 0.001$), and ns indicates no significant difference. Source data are provided as a Source Data file.

Fig. 5), which may be the reason for the significant up-regulation by drought.

## Functional divergence of *GhCIPK6D1* and *GhCIPK6D3* in cotton drought response

To elucidate the functions of *GhCIPK6D1* and *GhCIPK6D3* in cotton under drought stress, we generated overexpression lines (OE4, OE12) and knockout lines (*ko#5, ko#6*) of *GhCIPK6D1* (Supplementary Fig. 6), as well as overexpression lines (OE24, OE35) and RNAi lines (Ri16, Ri19) of *GhCIPK6D3*[38]. The transgenic and wild-type (WT) plants were grown under the same conditions and subjected to water deprivation at the four-leaf stage. The results showed that the mutant lines (*ko#5, ko#6*) of *GhCIPK6D1* exhibited a drought-resistant phenotype, while the OE4 and OE12 lines exhibited premature leaf wilting compared to the WT after 10 days of drought stress (Fig. 2a). Furthermore, we analyzed the content of malondialdehyde (MDA) and the transpiration rate (Tr) in plants. There were no significant differences between the WT and transgenic lines under normal conditions. However, under drought stress, the content of MDA and Tr were found to be significantly lower in *ko#5* and *ko#6* mutants, and significantly higher in OE lines than in WT (Fig. 2b, c). Meanwhile, other drought-related indicators, including proline, $H_2O_2$ and relative water content of leaves were also determined (Supplementary Fig. 7a–c), which revealed that mutation of *GhCIPK6D1* enhanced drought resistance. These results indicate that *GhCIPK6D1* negatively regulates drought resistance in cotton.

Interestingly, drought resistance effects of *GhCIPK6D3* showed an opposite trend to *GhCIPK6D1*. There were no significant differences between the WT and transgenic lines under well-watered conditions. However, compared to the WT, the *GhCIPK6D3*-OE lines (OE24, OE35) exhibited stronger drought resistance, while the RNAi lines Ri16 and Ri19 showed premature leaf wilting after 10 days of drought stress (Fig. 2d). The content of MDA and transpiration rate (Tr) of *GhCIPK6D3* transgenic plants were further analyzed. There were no significant differences between the WT and transgenic lines under watered conditions. However, the MDA and Tr of the OE lines were significantly lower than that of WT under drought stress. On the other hand, the Tr of the RNAi lines was significantly higher than that of WT (Fig. 2e, f). While the water use efficiency (WUE) of OE lines was higher than WT, with RNAi lines showed lower WUE (Supplementary Fig. 7d). We re-examined the expression levels of eight *GhCIPK6* homologous genes in *GhCIPK6D3*-RNAi lines, and found that RNAi lines exhibited down-regulated the expression of *GhCIPK6A1/D1/A3/D3/A4/D4*, with the expression of *GhCIPK6D3/A3* declining the most. While Ri16 showed increased expression of *GhCIPK6A2/D2*, Ri19 showed the opposite trend (Supplementary Fig. 8). These results suggest that *GhCIPK6D3* plays a positive regulatory role in drought stress response in cotton.

To investigate the functional differentiation of *GhCIPK6D1* and *GhCIPK6D3*, we captured images of stomata in overexpression and RNAi lines using scanning electron microscopy (SEM) and measured stomatal aperture using optical microscopy (Fig. 3). There was no significant difference between *GhCIPK6D1* transgenic plants and WT under watered conditions. However, the stomatal aperture of the knockout mutant lines was significantly lower than WT under drought stress, while the overexpression lines exhibited a significantly higher stomatal aperture compared to WT (Fig. 3a). However, a different scenario was observed for *GhCIPK6D3* transgenic plants. Under normal conditions, the stomatal aperture of the OE lines of *GhCIPK6D3* was significantly lower than that of WT, while the RNAi lines exhibited a significantly higher stomatal aperture compared to WT under watered conditions, as well as drought stress (Fig. 3b). The statistical analysis of stomatal aperture was consistent with phenotype (Fig. 3c, d). We also statistically analyzed the stomatal density under watered and drought stress, The results indicate that GhCIPK6D1 had no impact on stomatal density (Supplementary Fig. 9a), but OE-*GhCIPK6D3* decreased stomatal density, and its drought resistance might be related to this change of stomatal density (Supplementary Fig. 9b). These results indicate that, under drought stress, stomatal closure is promoted by reduced levels of *GhCIPK6D1* expression but by high levels of *GhCIPK6D3* expression (which also reduces stomatal density), illustrating the functional differentiation of *GhCIPK6D1* and *GhCIPK6D3*.

Stomatal movement often correlates with the $K^+$ current in guard cells. To verify whether the regulation of stomatal aperture by *GhCIPK6D1* and *GhCIPK6D3* in cotton is related to $K^+$ flux, Non-invasive Micro-test Technology (NMT) were used to test the direction and velocity of $K^+$ flux in the guard cells of transgenic and WT plants. It was found that the guard cells of mutant lines exhibited $K^+$ efflux (associated with stomatal closure), while *GhCIPK6D1*-OE lines exhibited $K^+$ influx under drought stress (associated with stomatal opening), while there was no difference on $K^+$ flux among *GhCIPK6D1* transgenic plants and WT under watered conditions (Fig. 3e). On the other hand, a different pattern of $K^+$ flux was observed in *GhCIPK6D3* transgenic plants. The guard cells exhibited high $K^+$ efflux (and stomatal closure) in OE lines and WT both under normal conditions and drought stress, while the increase of $K^+$ efflux in OE lines was higher than that in WT. However, the RNAi lines showed low $K^+$ flux (and stomatal opening), even under drought stress (Fig. 3f). These results indicate that *GhCIPK6D1* and *GhCIPK6D3* differentially regulate the current and velocity of $K^+$ flux in guard cells, thereby controlling stomatal aperture associated with differences in cotton drought tolerance.

## Polyploidization affects the interaction between CIPK6s and CBL1/2

CIPKs are activated after interacting with CBLs and participate in different stress responses. In view of the functional differentiation of GhCIPK6D1 and GhCIPK6D3, four *CBL1* and two *CBL2* genes were identified respectively from two diploid and upland cotton. To investigate the possible interactions between CIPK6 and CBL1/2 family members in diploid and upland cotton, yeast-2-hybrid experiments were carried out. The full-length coding sequences (CDS) of *CBL1/2* were cloned without mutation, and transformed into yeast Y187; while the full-length CDS of *CIPK6s* without mutation were transformed into yeast Y2H. Results show that each pair of combinations could grow normally and consistently on SD-L-T medium (Supplementary Fig. 10). However, significant differences were observed on SD-L-T-H medium.

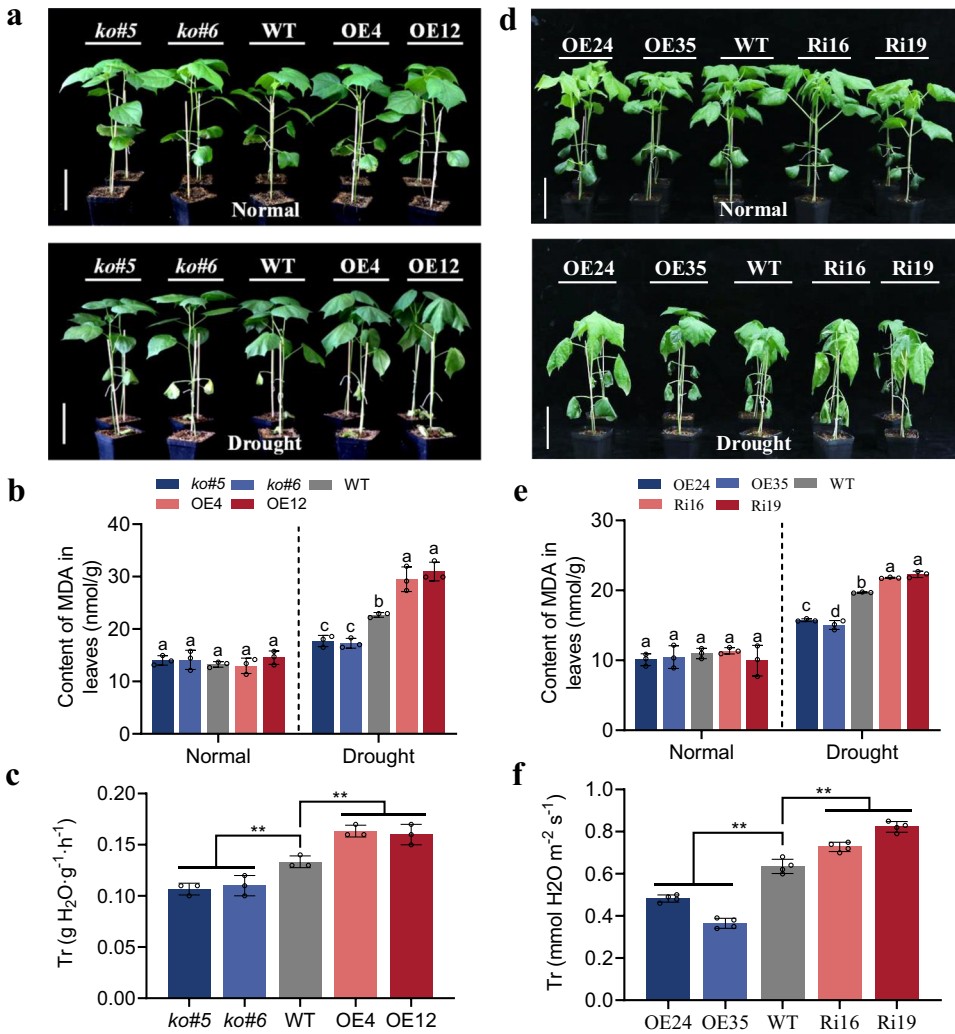

**Fig. 2 | *GhCIPK6D1* negatively and *GhCIPK6D3* positively regulates drought tolerance in cotton. a** and (**d**) Phenotypes of transgenic and WT plants under watered and drought stress treatments. Plants at four-leaf stage in soil were exposed to drought stress for 10 days. OE4/OE12 are overexpression lines and *ko#5/ ko#6* are knockout lines of *GhCIPK6D1*, OE24/OE35 are overexpression lines and Ri16/Ri19 are RNAi lines of *GhCIPK6D3*. Bars = 10 cm. **b** and (**e**) MDA content in the leaves of transgenic and WT lines under watered and drought stress. Data are means ± SD ($n$ = 3 biological replicates). Different letters above the columns of each compartment indicate a significant difference at $P < 0.05$ (one-way ANOVA followed by Duncan's multiple range test). **c** and **f** Transpiration rate in transgenic and WT lines under drought stress. Data are means ± SD ($n$ = 3) biological replicates in (**c**); $n$ = 4 biological replicates in (**f**). Significant difference analysis used one-way ANOVA followed by Duncan's multiple range test (**$P < 0.01$), and ns indicates no significant difference. Source data are provided as a Source Data file.

In *G. raimondii*, the interaction between 6D2/6D4 and CBL1/2 did not occur, but interaction was seen between 6D3 and CBL1/2 and appears to have arisen in the Dt subgenome of upland cotton after tetraploidization (Fig. 4a). When protein interactions were compared between candidates from *G. arboretum* and the At subgenome of upland cotton, the interaction between 6A1 and CBL1/2 was not detected, the interaction between 6A2/6A3 and CBL1/2 decreased, but there was interaction between 6A2 and CBL2 after tetraploidization (Fig. 4b). Interactions between CIPK6 and CBL1/2 can also be formed with proteins encoded across both Dt and At subgenomes of upland cotton. GhCIPK6D3 interacts with GhCBL1A1/2A1, and GhCIPK6A3 can interact with GhCBL1D1/2D1, but GhCIPK6D1 only interacts with GhCBL1A1 (Fig. 4c). The subcellular localization in tobacco and cotton protoplasts showed that GhCBL1A1 and GhCIPK6D1 were localized to plasma membrane and nucleus (Fig. 4d-e). The interaction between GhCBL1A1 and GhCIPK6D1 was further analyzed by BiFC in tobacco and cotton protoplasts (Fig. 4f, g). The results showed that both GhCBL1A1 and GhCIPK6D1 interacted at the plasma membrane and nucleus in tobacco (Fig. 4f), and displayed a high interaction signal at

the plasma membrane in cotton protoplasts (Fig. 4g). The pull-down assay also indicated that the interaction between GhCBL1A1 and GhCIPK6D1 in vitro (Fig. 4h). The interaction between GhCIPK6D3 and GhCBL2A1 in vitro and in vivo has been fully confirmed in our previous studies[38]. We re-performed experiments to investigate localization of GhCBL2A1 and interaction with GhCIPK6D3 by BiFC in cotton protoplasts. The results showed that GhCBL2A1 and GhCIPK6D3 indeed interact on the vacuole membrane (Supplementary Fig. 11). This suggests that *GhCIPK6* homologous genes showed high similarity but interacted with different *GhCBLs* and may therefore display different biological functions.

Amino acid and protein structure determines interaction. To explore the reason for the interaction difference between CIPK6s and CBL1/2 in *Gossypium*, we investigated the amino acid sequences, conserved motifs and 3D protein structure predictions. The corresponding CBL proteins were consistent in conservative domains before and after polyploidization. Compared with other CBLs, GhCBL1A1 lacks motif 6, which makes its 3D protein structure significantly different from other CBLs. GhCBL2A1/2D1 has an additional motif 7, which

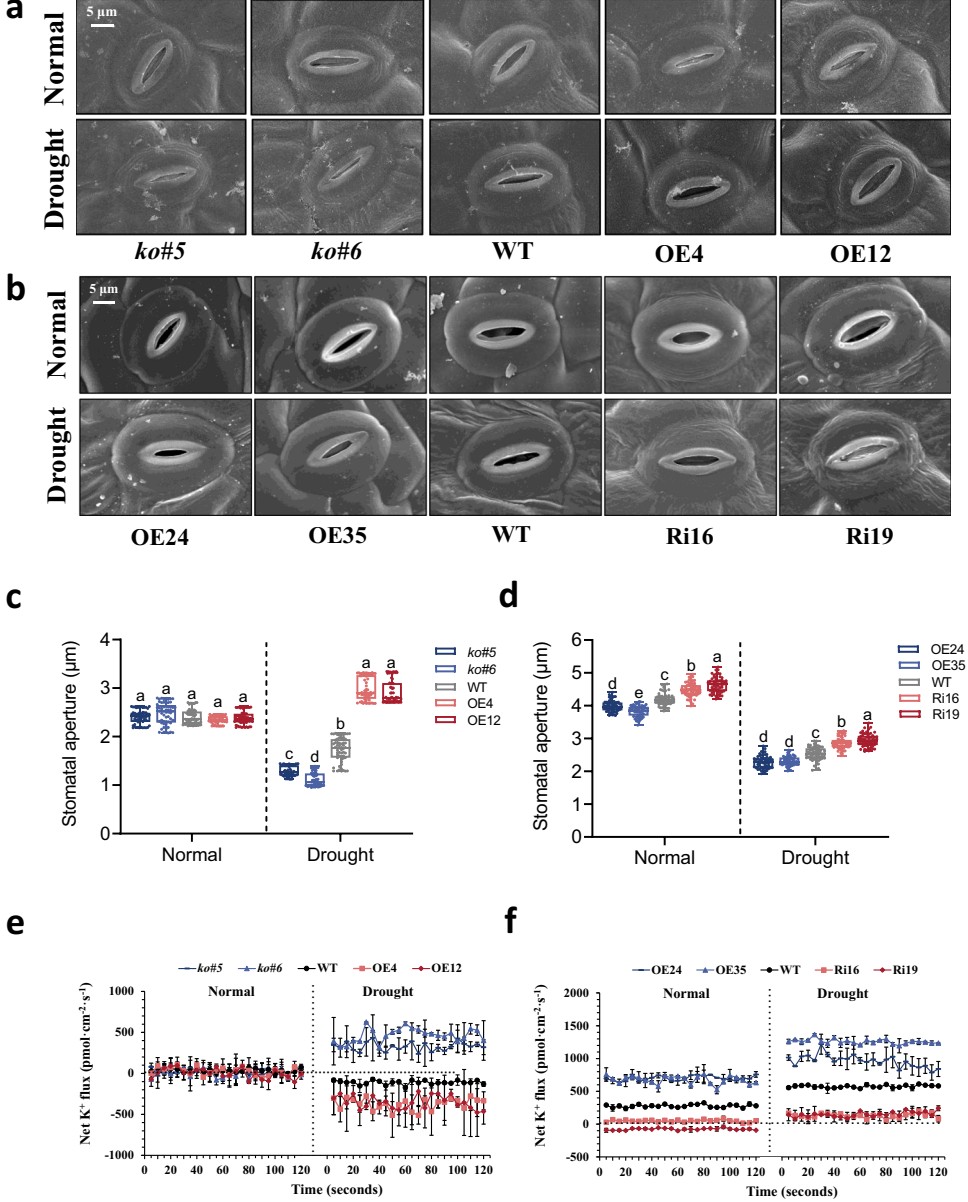

**Fig. 3 | *GhCIPK6D1* positively regulates stomatal opening but *GhCIPK6D3* positively regulates stomatal closure in cotton. a, b** Scanning electron microscopy images of leaf epidermal stomata of transgenic and WT lines under watered and drought stress. Bars = 5 μm. **c, d** Stomatal aperture in transgenic and WT lines under watered and drought stress (means ± SD, *n* = 60). Different letters above the boxes of each compartment indicate a significant difference at *P* < 0.05 (one-way ANOVA followed by Duncan's multiple range test). All box plots with centre lines showing the medians, boxes indicating the interquartile range, and whiskers indicating a range of minimum to maximum data beyond the box. **e, f** Measurement of K⁺ flux in guard cells of the second leaf of transgenic and WT lines under drought stress treatments. Data are means ± SD (*n* = 2) biological replicates. The average of stable 2 min data (*n* = 24) is selected. Source data are provided as a Source Data file.

might result in the observed tonoplast localization. GhCBL1D2 lacks motif 4, but has additional motif 9 and 10, though its 3D protein structure was unaffected (Supplementary Fig. 12a, b). Conversely, there was significant divergence in the CIPK6 amino acid sequences from diploid to tetraploid. For example, GhCIPK6A1/6A2/6D2/6D4 had increased or decreased fragmentation compared with the GaCIPK6s/GrCIPK6s, resulting in changes of 3D protein structure and a reduction in the numbers of interaction pairs after tetraploidization. This may be the reason why the interactions of 6A1/A2/D2/D4 decreased or disappeared and the interactions of 6D3 appeared after tetraploidization (Supplementary Fig. 13). In addition, we found numerous variations in the amino acid sequences of GhCIPK6D1 and GhCIPK6D3, which also led to differences in conserved protein motifs and protein 3D spatial configuration (Supplementary Fig. 14). GhCIPK6A3/D3 lack of motif 12

in contrast with other GhCIPK6s except for GhCIPK6D4, which may explain why only GhCIPK6A3/D3 could interact with GhCBL2A1/2D1.

### GhCIPK6D1 and GhCIPK6D3 reversely regulate drought resistance by forming GhCBL1A1-GhCIPK6D1 and GhCBL2A1-GhCIPK6D3 networks in cotton

As GhCIPK6D1 and GhCIPK6D3 were found to interact with GhCBL1A1, we generated overexpression lines (CO2, CO4) and knockout mutant lines (*cbl1-5*, *cbl1-13*) of *GhCBL1A1* to elucidate the function of *GhCBL1A1* in cotton under drought stress (Supplementary Fig. 15). The transgenic and WT plants were grown under the same conditions and subjected to water deprivation at the 4-leaf stage. The results reveal that the mutant lines of *GhCBL1A1* (*cbl1-5*, *cbl1-13*) were more resistant to drought, while the *GhCBL1A1*-OE lines (CO2, CO4) were more

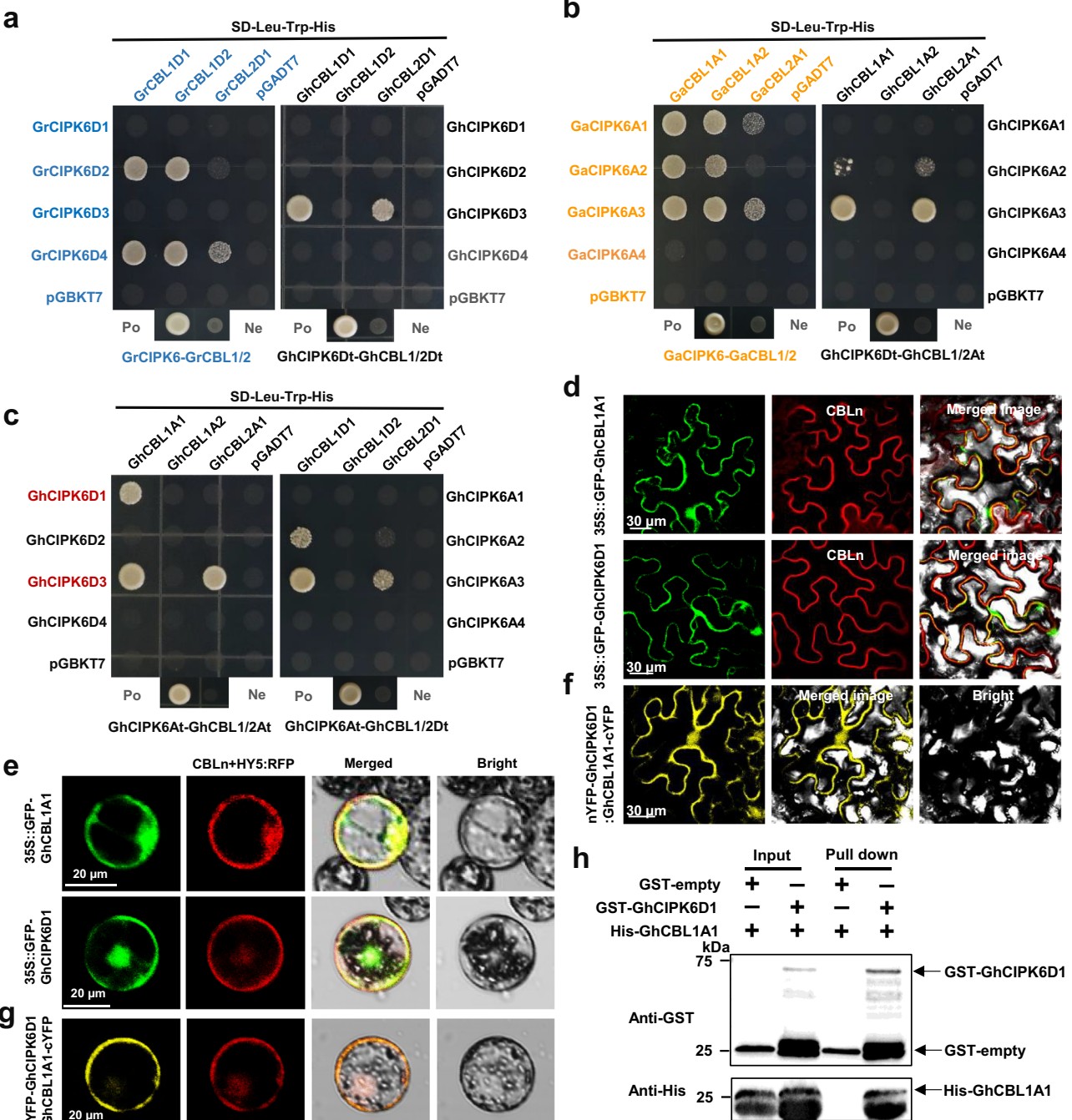

**Fig. 4 | Interaction pattern analysis between CIPK6 and CBL1/2 in *Gossypium*.** **a, b** Comparative analysis of interaction patterns between CIPK6 and CBL1/2 in *G. raimondii* and tetraploidized upland cotton Dt subgenome (**a**) and in *G. aboreum* and tetraploidized upland cotton At subgenome (**b**) on SD-Leu-Trp-His medium. **c,** Analysis of interaction patterns between GhCBL1/2 and GhCIPK6s in upland cotton on SD-Leu-Trp-His medium. "Po" represents positive control and "Ne" represents negative control. The positive control was the diploid hybrid yeast containing pGBKT7-53 and pGADT7-T, the negative control was the diploid hybrid yeast containing pGBKT7-Lam and pGADT7-T. **d, e** Subcellular localization of GhCBL1A1 and GhCIPK6D1 in tobacco. Bars = 30 μm (**d**). And in cotton protoplasts. Bars = 20 μm (**e**). **f, g** BiFC assays of GhCIPK6D1 and GhCBL1A1 in tobacco epidermal cells. Bars = 30 μm (**f**). And in cotton protoplasts. Bars = 20 μm (**g**). CBLn and HY5:RFP represent membrane and nucleus markers respectively. **h** Pull-down assay of GST-GhCIPK6D1 and His-GhCBL1A1. The arrows represent the target bands, the marker molecular weight is specified with number (kDa). The experiment was independently repeated three times (d-h). Source data are provided as a Source Data file.

sensitive to drought than WT after 10 days of drought stress (Fig. 5a). Furthermore, we analyzed the MDA and $H_2O_2$ contents (markers of drought stress response) of both WT and transgenic plants. Under watered conditions, no significant differences were observed between the WT and transgenic lines. However, under drought stress, the MDA and $H_2O_2$ contents in the *GhCBL1A1* knockout mutant lines were significantly lower than in WT, while the contents in the overexpression

lines were significantly higher than those in WT (Fig. 5b, Supplementary Fig. 16a). These results indicate that downregulation of *GhCBL1A1* is required to promote drought tolerance in cotton.

To further elucidate the role of *GhCBL1A1* in drought responses, we measured Tr and stomatal aperture using scanning electron and light microscopy. The results showed no significant difference between *GhCBL1A1* transgenic and WT plants under watered

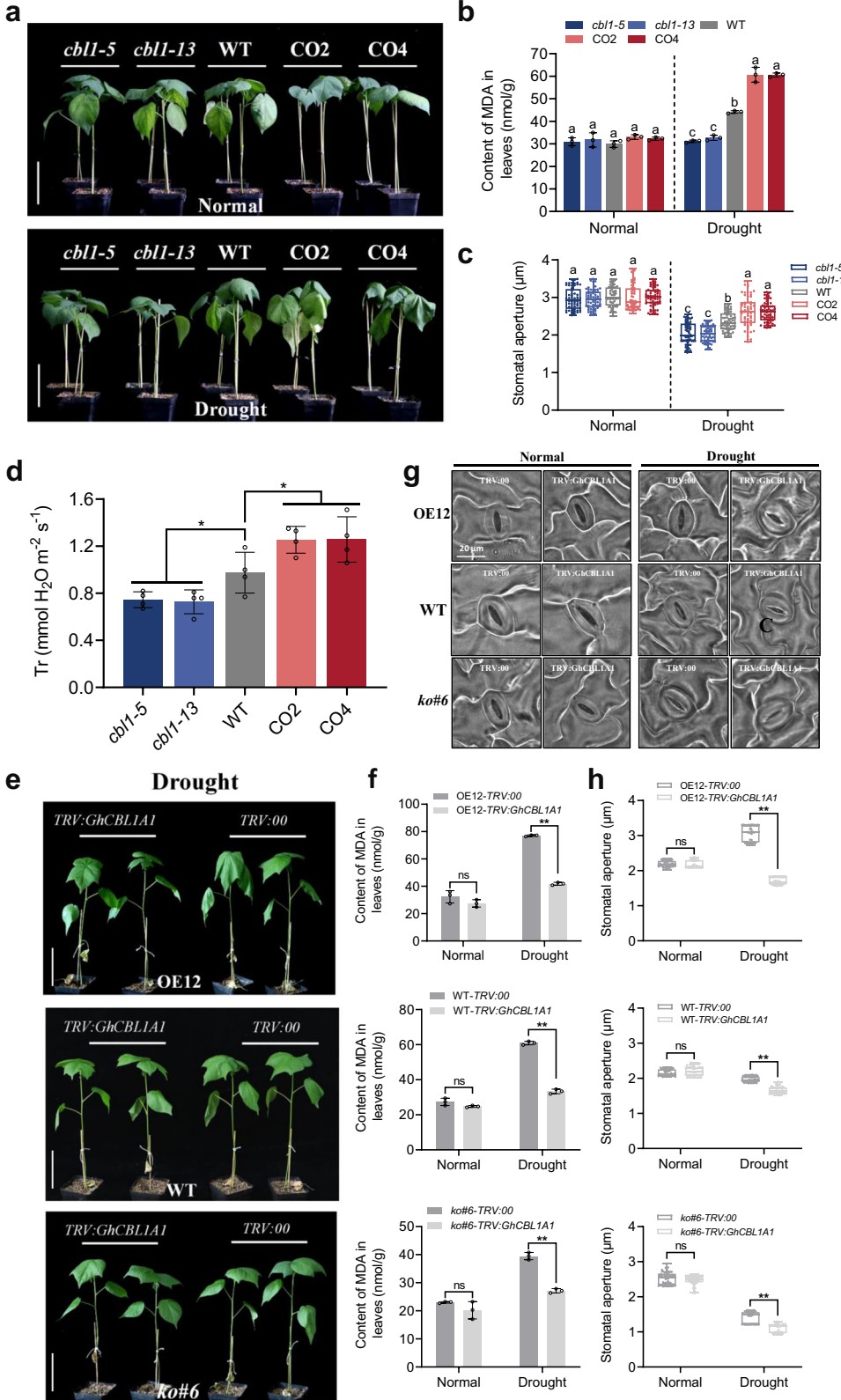

conditions. However, under drought stress, the stomatal aperture and Tr of the knockout mutant lines were significantly reduced compared to WT, while the overexpression lines exhibited a significantly larger stomatal aperture compared to WT (Fig. 5c-d, Supplementary Fig. 16b). There was no significant difference in stomatal density between transgenic plants and WT (Supplementary Fig. 16c). These results show that low levels of *GhCBL1A1* expression promote stomatal closure in

cotton under drought stress, associated with observed improved drought tolerance.

CBLs-CIPKs are involved in a wide range of plant stress responses, and in our study, the interaction between GhCIPK6D1 and GhCBL1A1 after tetraploidization was observed (Fig. 4). To elucidate whether *GhCBL1A1* and *GhCIPK6D1* have synergistic effects on cotton drought tolerance, we utilized Virus-induced gene silencing (VIGS) technology

**Fig. 5 | GhCBL1A1 and GhCIPK6D1 synergistically promote drought sensitivity of cotton. a** Phenotypes of WT and *GhCBL1A1* transgenic plants under watered and drought stress treatments. Plants at four-leaf stage in soil were exposed to drought stress for 10 days. CO2/CO4 were overexpression lines and *cbl1-5/cbl1-13* were knockout lines of *GhCBL1A1*. Bars = 10 cm. **b** MDA content in the leaves of transgenic and WT lines under watered and drought stress. Data are means ± SD (*n* = 3 biological replicates). Different letters above the columns of each compartment indicate a significant difference at *P* < 0.05 (one-way ANOVA followed by Duncan's multiple range test). **c** Stomatal aperture in transgenic and WT lines under normal and drought stress (means ± SD, *n* = 60). Different letters above the boxes of each compartment indicate a significant difference at *P* < 0.05 (one-way ANOVA followed by Duncan's multiple range test). **d** Transpiration rate in transgenic and WT lines under drought stress. Data are means ± SD (*n* = 4). Significant difference analysis used one-way ANOVA followed by Duncan's multiple range test (*\*P* < 0.05). **e** Phenotypes of *GhCBL1A1*-silenced plants in WT, OE12 and *ko#6*

backgrounds under drought stress. Plants at four-leaf stage in soil were exposed to drought stress for 10 days. Bars = 10 cm. **f** MDA content in the leaves of *GhCBL1A1*-silenced plants in WT, OE12 and *ko#6* backgrounds under normal watered and drought stress conditions. Data are means ± SD (*n* = 3 biological replicates). Significant difference analysis used two-tailed Student's *t*-test (*\*\*P* < 0.01), and ns indicates no significant difference. **g** Light microscopy images of leaf epidermal stomata of *GhCBL1A1*-silenced plants in WT, OE12 and *ko#6* backgrounds under normal watered and drought stress. Bars = 20 µm. **h** Stomatal aperture of *GhCBL1A1*-silenced plants in OE12, WT and *ko#6* backgrounds under. watered conditions and drought stress (means ± SD, *n* = 60). Significant difference analysis used two-tailed Student's *t*-test (*\*\*P* < 0.01), and ns indicates no significant difference. All box plots with centre lines showing the medians, boxes indicating the interquartile range, and whiskers indicating a range of minimum to maximum data beyond the box. Source data are provided as a Source Data file.

to knock down the expression of *GhCBL1A1* in OE12, WT and *ko#6* backgrounds of *GhCIPK6D1* (Supplementary Fig. 16d). Plants were subjected to water deprivation at the 4-leaf stage. The results show that after 10 days of drought treatment, silencing *GhCBL1A1* in OE12, WT and *ko#6* backgrounds resulted in better drought resistance compared to control *TRV:OO* plants (Fig. 5e). Measurement of MDA and proline (Pro) content in the leaves revealed that silencing *GhCBL1A1* in OE12, WT and *ko#6* backgrounds resulted in a significant decrease in MDA content and higher Pro content (Fig. 5f, Supplementary Fig. 16e). Furthermore, it was observed that the silencing of *GhCBL1A1* led to the closing of cotton stomata under drought. Under watered conditions, there was no significant difference in stomatal aperture between OE12, WT or *ko#6* backgrounds and *TRV:OO* plants. However, the stomatal aperture of *TRV: GhCBL1A1* plants was smaller compared to *TRV:OO* plants in drought treatment (Fig. 5g-h). These results suggest that GhCBL1A1-GhCIPK6D1 is part of the signaling system that positively regulates stomatal opening, resulting increased drought sensitivity in cotton; while downregulation of these genes leads to a synergistic promotion of stomatal closure and enhanced drought tolerance.

As GhCIPK6D3 was found to interact with GhCBL1A1 and GhCBL2A1 in Y2H, VIGS technology was used to suppress the expression of *GhCBL1A1* or *GhCBL2A1* in the *GhCIPK6D3* overexpression line OE24 (Supplementary Fig. 17a). The results showed that *GhCBL2A1* but not *GhCBL1A1* modulated the drought resistant effects of *GhCIPK6D3* through effects on leaf transpiration rate (Supplementary Fig. 17b) and electrical conductivity (Supplementary Fig. 17c). As GhCIPK6D3 interacts with GhCBL2A1 on the vacuolar membrane[38], we generated *GhCBL2A1* overexpression lines (CO7, CO13) and knockout mutant lines (*cbl2-1, cbl2-2*) to elucidate the functions of *GhCBL2A1* in cotton under drought stress[38]. Plant growth and stress conditions were the same as for *GhCBL1A1* transgenic plants. The results reveal that the growth of OE lines CO7 and CO13 were more tolerant to drought, while the mutant lines (*cbl2-1, cbl2-2*) exhibited premature leaf wilting after 10 days of drought stress (Fig. 6a). We analyzed the content of drought stress markers MDA and $H_2O_2$ in both WT and transgenic plants. Under watered conditions, no significant differences were observed between the WT and transgenic lines. However, under drought stress, the MDA and $H_2O_2$ contents in the *GhCBL2A1*-OE lines were significantly lower than in WT, while the knockout mutants were significantly higher than those in the WT (Fig. 6b-c). These results indicate that *GhCBL2A1* is a positive regulator of drought stress tolerance in cotton.

To further understand the basis for the positive effects of *GhCBL2A1* in cotton drought tolerance, we measured Tr and captured images of stomata and measured stomatal aperture using SEM and light microscopy. The results showed the stomatal aperture and Tr of the OE lines were significantly lower than that WT under drought stress, while the knockout mutants exhibited a significantly higher stomatal aperture compared to WT (Fig. 6d-f). It was also found that

overexpression of *GhCBL2A1* reduced stomatal density in cotton leaves, while there was no significant difference in stomatal density between mutants and WT (Supplementary Fig. 18). These results indicate that high levels of *GhCBL2A1* expression promote stomatal closure under drought stress and simultaneously reduce stomatal density to reduce water loss.

To determine whether GhCBL2A1 and GhCIPK6D3 act synergistically to promote cotton drought tolerance, we performed interspecific hybridization between the *GhCIPK6D3* overexpresser OE24 as one parental line and the *GhCBL2A1* mutant as the other, resulting in the generation of *GhCBL2A1* mutants OE24 × *cbl2-1* and OE24 × *cbl2-2* (i.e. *cbl2* mutants in the *GhCIPK6D3* overexpression background). Both transgenic and WT plants were grown under the same conditions and subjected to drought treatment at the 4-leaf stage. The results showed that after 10 days of drought stress, the OE24 lines grew normally compared to the WT, while the OE24 × *cbl2-1* and OE24 × *cbl2-2* mutants exhibited premature leaf wilting, and severe wilting was observed in the *cbl2-1* and *cbl2-2* mutants (Fig. 7a). Measurements of MDA and $H_2O_2$ content in the leaves indicated that under drought stress, the levels of each in the OE24 × *cbl2-1* and OE24 × *cbl2-2* lines were significantly higher than in OE24, and *cbl2-1* and *cbl2-2* mutants had significantly higher levels than the other genotypes (Fig. 7b). It was also found that mutation of *GhCBL2A1* in the *GhCIPK6D3* overexpression background affected stomatal aperture in cotton (Fig. 7c). Under normal growth conditions, stomatal apertures in the OE24 × *cbl2-1* and OE24 × *cbl2-2* lines were significantly larger than in OE24, while significantly smaller than in the *cbl2-1* and *cbl2-2* mutants, and a similar trend was observed under drought stress (Fig. 7d). Associated with the changes in stomatal aperture, the transpiration rate of the plants was correspondingly affected (Fig. 7e). These results further support the view that GhCBL2A1 and GhCIPK6D3 represent important components of the signaling system that positively regulates stomatal closure, so regulating drought tolerance in cotton.

## Discussion

Polyploidy plays a crucial role in enhancing the diversity of plant species and driving the evolution of genes, thus contributing to the formation of significant traits. Allotetraploid upland cotton (*G. hirsutum* L.), currently an extensively cultivated cultivar, originated ~1.5 million years ago through the interspecies hybridization of *G. arboretum* (AA) and *G. raimondii* (DD)[47,48]. Allopolyploidy serves as a vital mechanism for plants to adapt to environmental changes and evolution. The duplicated genes generated from polyploidization may exhibit different evolutionary fates, resulting in diverse functional categories. Here we report that two homologous genes, *GhCIPK6D1* and *GhCIPK6D3*, exhibit completely opposite regulation patterns of drought tolerance in cotton. As the first member of the eight homologous genes, *GhCIPK6D1* has a very low expression level in leaves in normal conditions, and its expression increases sharply when plants are subjected to drought stress. However,

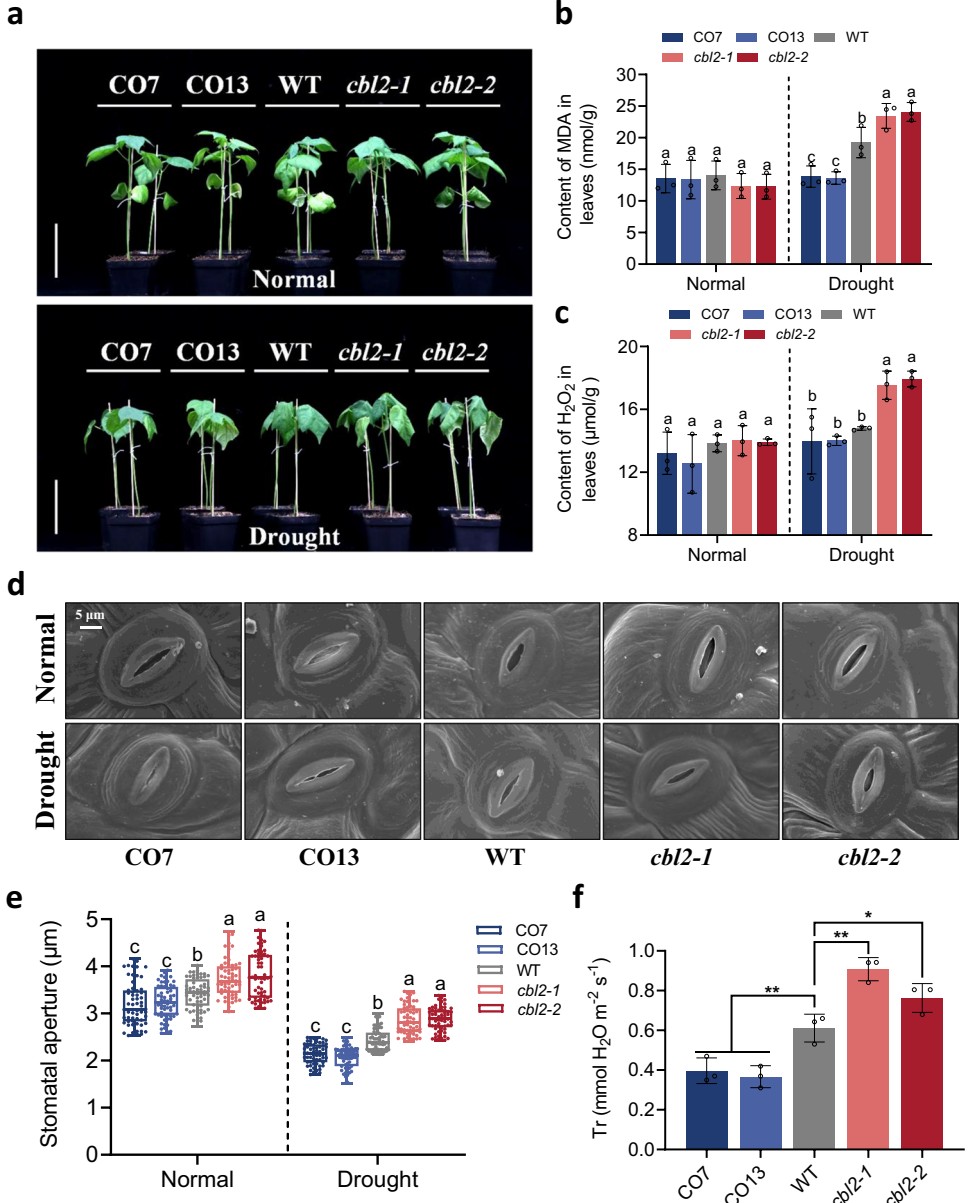

**Fig. 6 | GhCBL2A1 positively regulates drought tolerance in cotton.**
**a** Phenotypes of transgenic and WT plants under watered and drought stress treatments. Plants at four-leaf stage in soil were exposed to drought stress for 10 days. CO7/ CO13 are overexpression lines and *cbl2-1/cbl2-2* are knockout lines of *GhCBL2A1*. Bars = 10 cm. **b**, **c** MDA (**b**) and $H_2O_2$ (**c**) content in the leaves of transgenic and WT lines under watered and drought stress. Data are means ± SD ($n = 3$ biological replicates). Different letters above the columns of each compartment indicate a significant difference at $P < 0.05$ (one-way ANOVA followed by Duncan's multiple range test). **d** Light microscopy and scanning electron microscopy images of leaf epidermal stomata of transgenic and WT lines under normal and drought stress. Bars = 5 μm. **e** Stomatal aperture in transgenic and WT lines under watered and drought stress (means ± SD, $n = 60$). Different letters above the boxes of each compartment indicate a significant difference at $P < 0.05$ (one-way ANOVA followed by Duncan's multiple range test). All box plots with centre lines showing the medians, boxes indicating the interquartile range, and whiskers indicating a range of minimum to maximum data beyond the box. **f** Transpiration rate in transgenic and WT lines under drought stress. Data are means ± SD ($n = 3$ biological replicates). Significant difference analysis used one-way ANOVA followed by Duncan's multiple range test (*$P < 0.05$, **$P < 0.01$). Source data are provided as a Source Data file.

the homologous gene *GhCIPK6D3*, which was subsequently duplicated, shows expression levels hundreds of times higher than that of *GhCIPK6D1* under watered conditions, and it was also significantly up-regulated during drought stress. Our results show that high levels of *GhCIPK6D1* expression promote drought sensitivity in cotton, while high levels of *GhCIPK6D3* expression promotes drought tolerance.

Moreover, we found that these two genes regulate the drought response by their proteins interacting with two calcineurin B-like proteins, GhCBL1A1 and GhCBL2A1 respectively, to modulate stomatal guard cell movement. Expression of each of the four genes is very high in guard cells (Supplementary Fig. 19a, b), accounting for between 62% and 74% of the total gene expression level in the leaf (Supplementary Fig. 19c). These results show that different members of cotton gene families have evolved different functions that allow adaptation to environmental change following polyploidy.

The CBL-CIPK signaling pathway is an important mechanism for decoding and transmitting calcium signals triggered by environmental stress, conferring diversity, specificity, and complexity to molecular

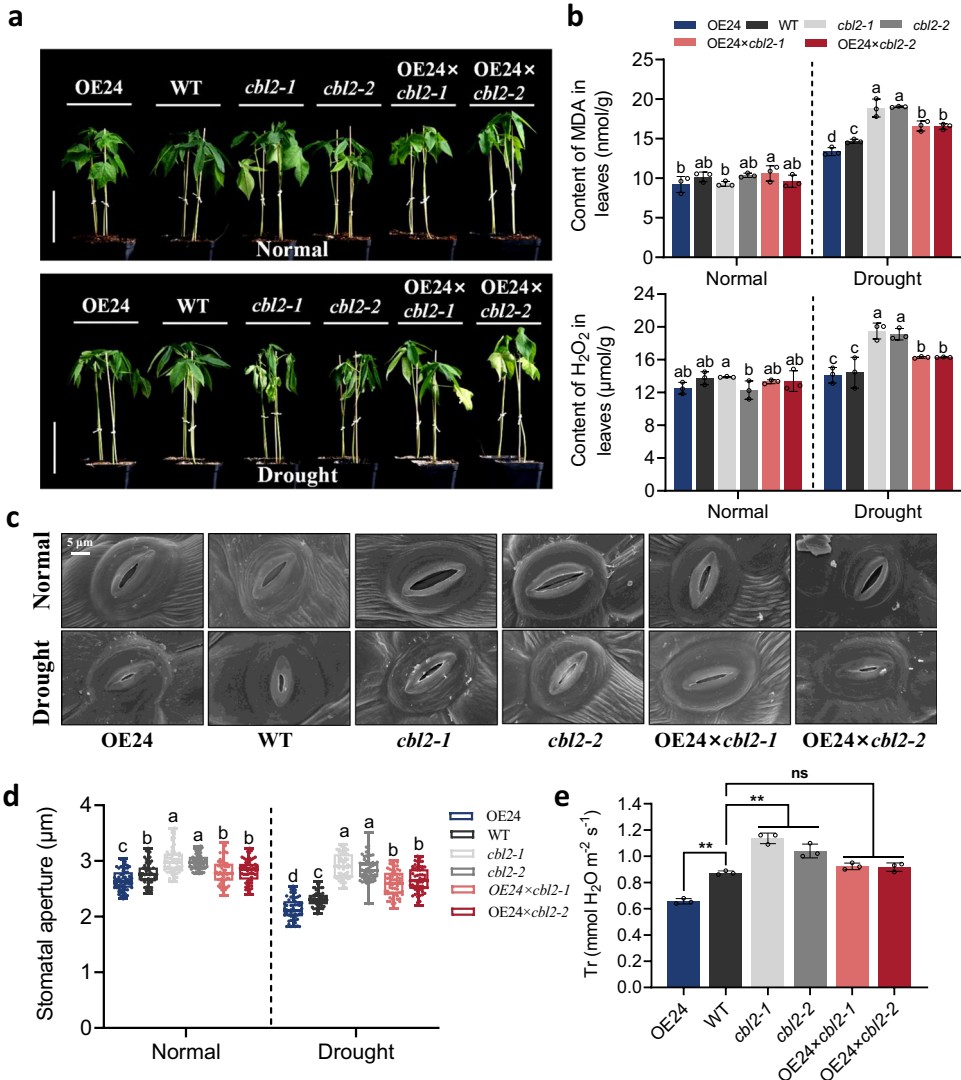

**Fig. 7 | The role of GhCIPK6D3 in drought response depends on GhCBL2A1.**
**a** Phenotypes of *GhCBL2A1* mutant plants in OE24 backgrounds under drought stress. Plants at four-leaf stage in soil were exposed to drought stress for 10 days. Bars = 10 cm. **b** MDA and $H_2O_2$ content in the leaves of transgenic and WT lines under normal watered and drought stress. Data are means ± SD ($n$ = 3 biological replicates). Different letters above the columns of each compartment indicate a significant difference at $P < 0.05$ (one-way ANOVA followed by Duncan's multiple range test). **c** Scanning electron microscopy images of leaf epidermal stomata of transgenic and WT lines under watered conditions and drought stress. Bars = 5 μm. **d** Stomatal aperture in transgenic and WT lines under watered conditions and

drought stress (means ± SD, $n$ = 60). Different letters above the boxes of each compartment indicate a significant difference at $P < 0.05$ (one-way ANOVA followed by Duncan's multiple range test). All box plots with centre lines showing the medians, boxes indicating the interquartile range, and whiskers indicating a range of minimum to maximum data beyond the box. **e** Transpiration rate in transgenic and WT lines under drought stress. Data are means ± SD ($n$ = 3 biological replicates). Significant difference analysis used one-way ANOVA followed by Duncan's multiple range test (**$P < 0.01$), and ns indicates no significant difference. Source data are provided as a Source Data file.

responses[23,49]. During evolution, the interaction pattern between *CIPK6* and *CBL1* or *CBL2* in cotton has changed. In the At subgenome, the logarithm of interactions decreases after tetraploidization, while in the Dt subgenome, the interaction between GhCIPK6D2/GhCIPK6D4 and GhCBL1 or GhCBL2 is no longer seen after tetraploidization, and another interaction, between GhCIPK6D3 and GhCBL1 or GhCBL2, emerges. Structural analysis reveals significant amino acid sequence evolution and protein structure changes in *CIPK6* following polyploidization, which may account for the decreased interactions in tetraploids (Supplementary Figs. 13–14).

Although yeast two-hybrid experiments have shown that GhCIPK6D1 interacts only with GhCBL1A1, GhCIPK6D3 can interact with both GhCBL1A1 and GhCBL2A1 (Fig. 4), but the regulation of GhCIPK6D3 is dependent on GhCBL2A1 and not on GhCBL1A1 (Supplementary Fig. 17). As calcium ion sensors, CBLs are widely involved in

stress responses[49]. Further validation of roles for *GhCBL1A1* and *GhCBL2A1* shows that under drought stress, *GhCBL1A1* positively regulates stomatal opening in cotton, while *GhCBL2A1* promotes stomatal closure, resulting in differences in drought tolerance between *GhCBL1A1* and *GhCBL2A1* transgenic lines, consistent with the functions of their respective interacting GhCIPK6s (Figs. 5–6). CBL-CIPK proteins often work synergistically to exert their functions[50]. GhCBL1A1 and GhCIPK6D1 mainly interact at the plasma membrane, while GhCBL2A1 and GhCIPK6D3 interact at the vacuolar membrane, suggesting that the potential phosphorylation targets of GhCIPK6D1 and GhCIPK6D3 are different, potentially an unidentified potassium transporter. In tetraploid upland cotton, both GhCIPK6D1 and GhCIPK6D3 form regulatory networks through interaction with CBLs encoded on the At subgenome, which may indicate that the fusion of different subgenomes after polyploidy is more conducive to the

formation of functional networks adapted to environmental stress under selection pressure. Given the antagonistic relationship between the GhCBL1A1-GhCIPK6D1- and GhCBL2A1-GhCIPK6D3-dependent networks in regulating stomatal aperture, we speculate that the different complexes differentially regulate downstream signaling networks to control guard cell activity. This information provides different insights into the evolution and interaction between CBL-CIPK and stomatal movement in response to drought stress as a consequence of genome duplication.

Stomatal movement is closely related to the flow direction of $K^+$ ions[51]. Our results showed that overexpression of *GhCIPK6D1* under drought stress resulted in influx of $K^+$ ions in leaf guard cells, which increased stomatal opening to increase plant sensitivity to drought, while efflux of $K^+$ ions in mutants guard cells reduces stomatal aperture, reduced water loss and promotes drought tolerance (Figs. 2 and 3). However, on overexpression of *GhCIPK6D3*, the guard cells showed increased $K^+$ efflux, a reduced stomatal opening and enhanced drought tolerance, while the RNAi lines showed a stronger $K^+$ influx, increased stomatal opening and increased drought sensitivity (Figs. 2 and 3). Potassium is essential for plant growth and development[52]. Since the regulation of the flow of $K^+$ in guard cells mediated by GhCIPK6D3 is the same under watered and drought conditions, GhCIPK6D3 may regulate cotton drought tolerance through other mechanisms than through $K^+$ flux control, and this requires further research. Nevertheless, *GhCIPK6D1* and *GhCIPK6D3* play an important role in regulating $K^+$ ion transport in the guard cells to regulate stomatal opening and closing (Fig. 8). Stomatal density is also closely related to plant drought tolerance. Our results show that *GhCIPK6D1* did not affect stomatal density, but the overexpression of *GhCIPK6D3* reduced stomatal density, which may also be caused by small differences in the morphology of transgenic plants. Therefore, drought tolerance mediated by *GhCIPK6D3* was a combination of stomatal movement and stomatal density effects.

The mechanism of stomatal movement has long been a focal point of research in plant physiology and cell signaling mechanisms. There are three main areas of study regarding the regulation of stomatal movement. In addition to $K^+$ regulating stomatal opening and closing, there is also starch-sugar conversion and a role for malic acid in this process. Our preliminary findings suggested that *GhTST2* is one of the downstream targets of *GhCIPK6D3*. As a vacuolar sugar transporter, *GhTST2* is involved in regulating the sugar content within cells. *GhCIPK6D3* regulates *GhTST2* to mediate sugar transport, which may have an impact on the opening and closing of stomata, and this could be a future research direction.

## Methods

### Plant materials, growth conditions and drought treatments
Seeds of cotton (*Gossypium hirsutum* cv. Jin668 and YZ1) were used for genetic transformation and VIGS assays. The cotton plants were cultivated in controlled environment rooms under conditions of 25℃ and a 16 h light/8 h dark photoperiod. Small pots filled with sterilized soil (humus: vermiculite, 2:1) were used for plant growth. Drought treatment was initiated at the four-leaf stage and lasted for 14 days, during which plants were exposed to natural drought.

### Genome data, phylogenetic analysis and orthogroup identification
A total of 23 sequenced genomes, including 2 diploid cotton (*G. raimondii* and *G. arboreum*) and 1 tetraploid cotton (*G. hirsutum*) were downloaded for phylogenetic analysis and orthogroup identification (Supplementary Data 1). Gene, CDS and promoter sequences of the *CIPK6* genes are provided in Supplementary Data 2.

The phylogenetic tree of 25 plant species (except *G. hirsutum*) was presented using the TimeTree Website (http://timetree.org/). The timing of WGDs were collected from the literature[53], and marked on the species tree. OrthoFinder (v2.5.4)[54], with default parameters, was used to identify *CIPK6* gene family members among 26 plant species, and the orthogroup with *Arabidopsis* gene *CIPK6* was identified as a *CIPK6* gene family. The linkage plot between 2 diploid cotton and 1 tetraploid cotton was performed by TBtools (v1.123)[55].

Four tetraploid cottons (*G. barbadense*, *G. tomentosum*, *G. mustelinum* and *G. darwinii*) were used to retrieve *CIPK6* genes from available genome data downloaded in *CottonMD* (https://yanglab.hzau.edu.cn/CottonMD)[56] by querying each *G. hirsutum* GhCBL1/2 and GhCIPK6s amino acid sequence. The *Gossypium* phylogenetic trees were constructed using the MAGA7.0 software by Neighbor-Joining method with 1000 bootstrap replicates. The evolutionary distances were determined using the Poisson correction method[57].

### Identification of gene duplications
The different modes of gene duplication were identified using the DupGen_finder pipeline (https://github.com/qiao-xin/DupGen_finder)[53] The genomes of *Selaginella moellendorffii*, *Amborella trichopoda*, *Spirodela polyrhiza*, *Setaria italic* and *Boechera stricta* were filtered due to lack of chromosomal level genome information. The genomes of *Musa acuminate* and *Hordeum vulgare* were filtered due to lack of gff annotations. *Physcomitrella patens* was used as outgroup in DupGen_finder analysis. Finally, the first peptide of all genes in 16 plant genomes were retained to perform DupGen_finder_unique.pl analysis with default paraments. The genes were classed into 5 modes: whole-genome duplication, tandem duplication, proximal duplication, transposed duplication and dispersed duplication.

### Ka/Ks estimation of *CIPK6s* gene
The ratio of nonsynonymous substitutions per nonsynonymous site (Ka) to synonymous substitutions per synonymous site (Ks) was calculated by the Simple Ka/Ks Calculator procedure in software TBtools[55]. According to the definition of Ka/Ks, values less than 1 represent negative or purifying selection, while values greater than 1 represent positive selection.

### Amino acid sequence alignment, conserved motifs and 3D protein structure prediction
The amino acid sequences of all CIPK6s were first compared by MEGA 7, then the comparisons were saved as *.msf files and embellished by GeneDoc tool. The amino acid sequences of all CBL1/2 and CIPK6s were used for prediction of motifs using the Multiple Em for Motif Elicitation (MEME) program (http://meme-suite.org/tools/meme). The optimum width of motifs was set to range from 4-33, the maximum number of motifs of CIPK6s is 16, CBLs is 10, and default values for other parameters. All amino acid sequences of CBL1/2 and CIPK6s were predicted and compared in 3D protein structure by Protein Homology/analogy Recognition Engine V 2.0 (Phyre[2]) software[58].

### Analysis of *cis*-acting elements in *CIPK6* promoters
The promoter sequences of the *CIPK6* genes (2000 bp upstream of the ATG initiation codon) were obtained from the cotton genome CottonFGD (https://cottonfgd.org/) and *CottonMD* (https://yanglab.hzau.edu.cn/CottonMD). PlantCARE (http://bioinformatics.psb.ugent.be/webtools/plantcare/html/search_CARE.html) was used to obtained the cis-acting elements of *CIPK6* promoter sequences[59]. Abiotic stress responsive elements and drought related elements were analyzed by TBtools for mapping[55]. All the cis-elements of *CIPK6s* in *Gossypium* are listed in Supplementary Data 3.

### Plant materials, vector construction, genetic transformation and drought treatments
Upland cotton *G. hirsutum* cultivar Jin668 and YZ1 were used in this study. Seeds of Jin668 were used for genetic transformation and VIGS assays. The overexpression constructs for *GhCBL1A1* and *GhCIPK6D1*

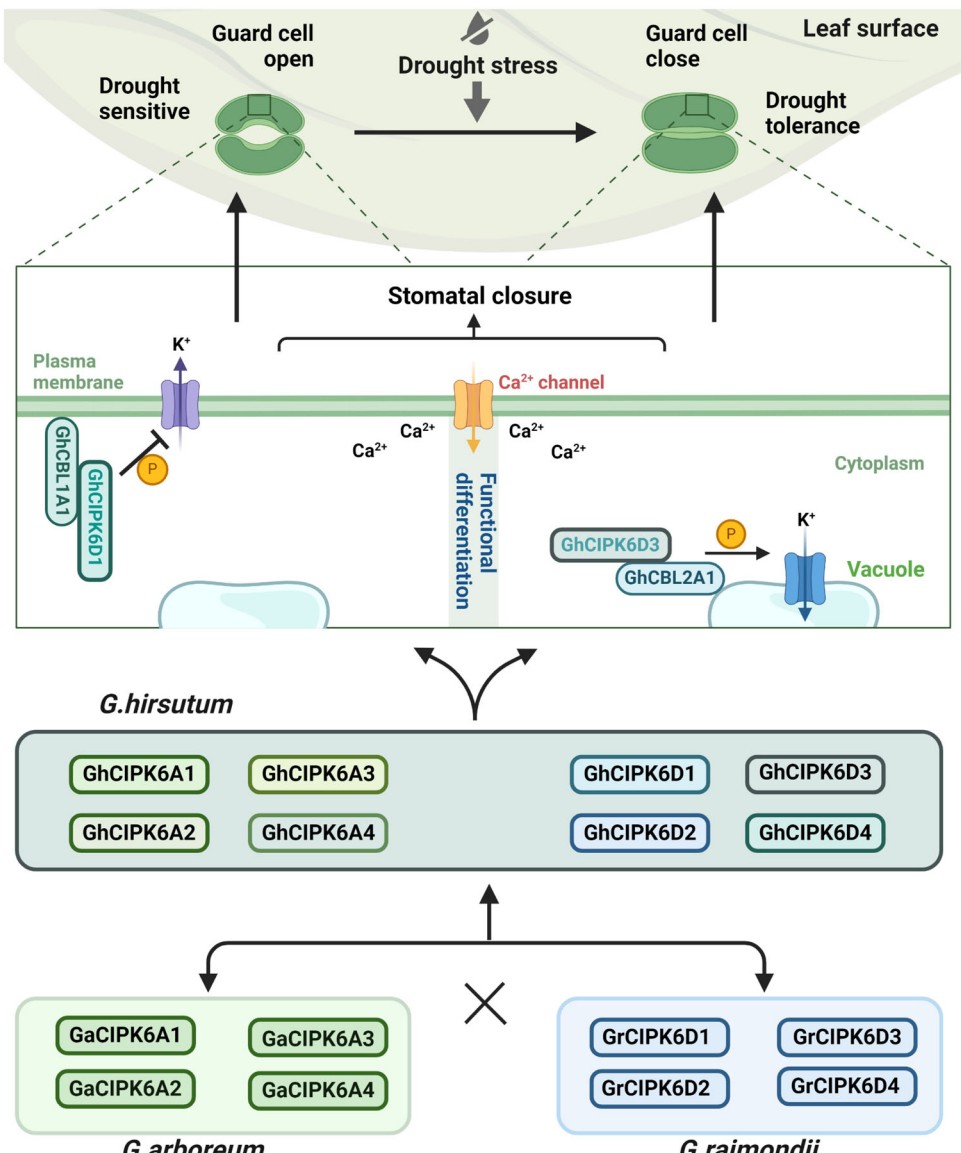

**Fig. 8 | Functional differentiation of GhCIPK6D1 and GhCIPK6D3 in regulating drought resistance of cotton.** After two WGDs, four *CIPK6* genes remained in *G. raimondii* and *G. arboretum*, after quadrupling, eight *CIPK6* genes were formed in upland cotton (*G. hirsutum*), and the expression of *GhCIPK6D1* and *GhCIPK6D3* and their biological functions in response to drought were sub-functionalized. Although the basal expression of *GhCIPK6D1* was very low, it was greatly up-regulated by drought stress, and its interacting protein GhCBL1A1 was also up-regulated. In overexpression plants (GhCBL1A1-GhCIPK6D1), the downstream target protein in the cell membrane may be phosphorylated by GhCIPK6D1, which inhibits guard cell K⁺ efflux, increases the stomatal opening of leaves, accelerates the water loss, and plants exhibit drought sensitivity. However, the basal expression of *GhCIPK6D3* was high, and also up-regulated by drought, and its interacting protein gene *GhCBL2A1* was also up-regulated. In overexpression plants (GhCBL2A1-GhCIPK6D3), the downstream target protein in the vacuole membrane might was phosphorylated by GhCIPK6D3, which influences the K⁺ flux, resulting in closure of guard cells, and plants exhibit drought tolerance. *GhCIPK6D1* is the likely progenitor of *GhCIPK6D3* which evolved from *GhCIPK6D1*. These results show that different cotton family genes have evolved subfunctions to adapt to environmental changes. The figure Created with BioRender.com released under a Creative Commons Attribution-NonCommercial-NoDerivs 4.0 International license.

were in the vector pK2GW7[60], and the CRISPR-Cas9 mediated gene editing vector was constructed by seamless cloning technology[61]. Transgenic plants were created by *Agrobacterium*-mediated transformation[62]. The expression levels of *GhCBL1A1* and *GhCIPK6D1* in transgenic plants were determined by qRT-PCR, using *GhUB7* as an internal control. The mutation analysis of *GhCBL1A1* and *GhCIPK6D1* CRISPR-Cas9 transgenic plants was achieved by extracting the genomic DNA and performing Hi-TOM sequencing[63], and we chose the mutants *Ghcbl1a1* and *Ghcipk6d1* carrying frameshift and marker-free mutations for phenotypic analysis. The transgenic plants for *GhCBL2A1* and *GhCIPK6D3* were used from our previous study[38]. Cotton plants were grown in culture rooms at 25°C with 6 h light/8 h dark photoperiod, and grown in nutrient soil and Hoagland solution. Four-leaf stage plants were used to perform drought treatment with natural drought. All the primers used for vector construction are listed in Supplementary Data 4.

### *CIPK6* expression analysis

The expression levels of *CIPK6s* in *G. hirsutum*, *G. raimondii* and *Gossypium arboreum* were determined by qRT-PCR under watered and drought conditions, using *GhUB7* as an internal control. Plants of the diploid and upland cottons were grown in the same light culture room (25°C with 16 h light/8 h dark photoperiod, and grown in nutrient soil) at the same time, until the plants grew to the 4-leaf stage, subjected to

drought treatment, and then sampled before and after drought to analyze gene expression. Tissue samples were collected and immediately frozen in liquid nitrogen and stored at −80 °C. The total RNA extraction method followed the same procedure as previously used in our lab[64]. In summary, the samples were initially ground into a powder using liquid nitrogen, and then 0.1 g of the powder was weighed and placed into a 2 ml centrifuge tube. The RNA was cleaved by adding sodium acetate and RNA lysate, followed by the addition of chloroform and centrifugation. Isopropyl alcohol was then added to facilitate RNA deposition, which was followed by two washes with 75% ethanol. Finally, the RNA was dissolved in nuclease-free water. RNA was reverse transcribed to cDNA using SuperScript III Reverse Transcriptase, according to the manufacturer's instructions. qRT-PCR analysis of selected *GhCBLs* and *GhCIPKs* was performed using an ABI Prism 7500 Real Time PCR system (ABI, Foster City, CA, USA). Relative expression values were calculated using the $2^{-\Delta\Delta CT}$ methods. The primers used for qRT-PCR are listed in Supplementary Data 4.

### Virus-induced gene silencing assays

The gene-specific fragment (300-500 bp) from the CDS region of *GhCBL1A1* was inserted into *pTRV2*. The empty vector *pTRV2* was used as negative control. The vector constructs were introduced into *Agrobacterium tumefaciens* strain GV3101 by electroporation. The cultures were adjusted to $OD_{600} = 0.8$ and samples of *A. tumefaciens* with *TRV1* vectors were mixed in equal volumes and infiltrated into cotyledons of transgenic plants OE12 (overexpression of *GhCIPK6D1*) and ko#6 (mutant of *GhCIPK6D1*) by vacuum infiltration. The seedings were grown in a culture room at temperature of 25°C with a 6 h light/8 h dark photoperiod (120 μmol m$^{-2}$ s$^{-1}$). The expression level of target genes was detected 2 weeks after infection. The successfully silenced plants were used for drought stress treatments and functional studies. The primers used in the VIGS assays are listed in Supplementary Data 4.

### Physiological analysis of leaves under drought stress

To investigate the changes in physiological indicators under drought stress, we subjected WT and transgenic plants or VIGS plants to drought treatment at the four-leaf stage. To determine the transpiration rate (Tr), the lower part of the plant was sealed with a plastic film to prevent soil moisture evaporation. The total weight of the plants and the nutrient pots was measured every day at 9:00 am, and the amount of water consumed by transpiration was calculated. After the appearance of the drought phenotype, the leaves were detached and photographed. The leaf area was calculated by Digimizer software (v5.4.4, MedCalc Software, Belgium). Tr was calculated as the amount of water consumed per unit leaf area per unit time.

To determine water use efficiency, each bowl was weighed and water was added appropriately to ensure the same mass. The nutrient pots were placed with a plastic film covering, and the plants were grown in the culture room until wilting occurred. The final weight of the nutrient pots was measured, and the total water consumption was calculated. The aboveground portion of the plants was sampled and dried to a constant weight, and the accumulated dry matter was measured. Water use efficiency was expressed as the accumulation of dry matter per unit of water consumption.

After 10 days of drought stress, the contents of MDA, $H_2O_2$, and Pro in the leaves were measured. Untreated leaves were used as the control, and all measurements were repeated three times. The contents of MDA, $H_2O_2$, and Pro were determined according to the instructions provided by Suzhou Graces Biotechnology Co., Ltd. (Suzhou, China), using the reagent kits they produce.

Stomatal phenotypes of plants were observed by scanning electron microscope (SEM) and light microscopy. A nail polish imprinting method was used for light microscope observation. Cotton plants subjected to 10 days of either drought stress or watered conditions for collection of the lower epidermis of the second true leaf of the plant. Transparent nail polish was evenly applied on the back of cotton leaves where there were no veins. The polish was allowed to dry naturally for 5–10 min, then scotch tape was pressed onto the dried nail polish coating and then gently removed. The tape with nail polish was transferred to a microscope slide, and stomatal imaging was observed using light microscopy (Zeiss Axio Scop A1, Oberkochen, Germany). For SEM observation, the cotton leaves were cut with scissors (4–5 mm$^{-2}$ in size) and immediately transferred to fixed solution (2.5% glutaraldehyde solution prepared with 0.2 M phosphoric acid buffer, pH 7.2). The samples were then vacuumed for 30 min, dehydrated using different concentrations of ethanol solutions, and then replaced with isoamyl acetate for 20 min. After critical point drying, gold spraying was performed. The stomata were observed and photographed using scanning electron microscopy (JSM-6390/LV SEM, Jeol, Tokyo, Japan). Digitizer (v5.4.4) software was used to calculate the stomatal aperture and number of pores. At least 60 stomata were analyzed and quantified per sample.

### K$^+$ flux measurement

Net fluxes of K$^+$ were measured using Non-invasive Micro-test Technology (NMT)[65]. To measure the K$^+$ flux, cotton plants subjected to 10 days of drought stress were used, with watered plants serving as a control. K$^+$ flux was measured using a non-destructive micrometer (NMT150-SIM-XY; Zhongguancun Alliance-Xuyue Company). Lower epidermis of the second leaf was carefully peeled off to expose the stomata. After equilibrium in K$^+$ testing solution (0.1 mM KCl, 0.1 mM $CaCl_2$, pH 6.0) for 5 min, the K$^+$ flux of guard cells was measured. At least 3 plants were measured, and at least 2–3 leaves were taken for each plant, with 2–3 stomata in different areas for measurement. Each measurement lasted at least 5 min, and stable data from the last 2 min were analyzed.

### Yeast two-hybrid screening

The Matchmaker Gold Yeast Two-Hybrid system (Cat. No. 630489) was used in Y2H assays. Full-length coding sequences of *CIPK6s* (*G. hirsutum*, *G. raimondii* and *G. arboreum*) were cloned into vector pGBKT7 and transformed into yeast strain Y2H Gold. Full-length sequences of *CBL1/2* (*G. hirsutum*, *G. raimondii* and *G. arboreum*) were constructed in vector pGADT7 and transformed into yeast strain Y187. Yeast two-hybrid screening was performed according to the manufacturer's instructions (Clontech, PT1172-1, PT4084-1). The positive transformants of bait and prey were mixed and mated at 30 °C for 72 h, and independent colonies of the same size transferred to SD -Leu-Trp medium. Interactions between different proteins were determined by growth on SD medium, SD -Leu-Trp and SD-Leu-Trp-His, respectively. The primers used are listed in Supplementary Data 4.

### Subcellular localization, BiFC assay and protoplast transformation

For subcellular localization assays, full length *GhCBL1A1*, *GhCIPK6D1* and *GhCBL2A1* sequences were cloned into the vector pMDC43 with GFP fused to the N-terminus by Gateway cloning technology[66]. For bimolecular fluorescence complementation (BiFC) assays, the full-length sequences of *GhCBL1A1* and *GhCIPK6D1 GhCBL2A1* and *GhCIPK6D3* were cloned into vector pDONR221 (Invitrogen™), then into the destination vector pBiFCt-2in1-NC by Gateway technology[67]. The recombinant vectors (*35S$_{pro}$*::GFP-GhCBL1A1, *35S$_{pro}$*::GFP-GhCIPK6D1 and nYFP-GhCIPK6D1:GhCBL1A1-cYFP) were transformed into *Agrobacterium tumefaciens* strain GV3101, and transient expression was performed in tobacco leaf epidermal cells. GFP fluorescence and YFP fluorescence in BiFC assays were detected under a confocal microscope (Olympus FV1200) 2d after Agrobacterium transfection. CBLn represents membrane markers[68]. The primers used are listed in Supplementary Data 4.

For the subcellular localization and BiFC assays in cotton protoplasts, the vectors fused with GFP and YFP (*35S*$_{pro}$::GFP-GhCBL1A1, *35S*$_{pro}$::GFP-GhCIPK6D1, nYFP-GhCIPK6D1:GhCBL1A1:cYFP, *35S*$_{pro}$::GFP-GhCBL2A1, nYFP-GhCIPK6D3:nYFP-GhCBL2A1) and the markers of CBLn:RFP, HY5:RFP and RabG3b:RFP were co-transformed into the prepared protoplasts using 40% polyethylene glycol 4000 (v/v) (Sigma), then cultured at 25 °C in darkness for 16 h. CBLn and HY5 represent membrane and nucleus markers respectively[68,69]. RabG3b was reported to be a vacuole membrane-localized protein[70], so RabG3b:RFP as a vacuole membrane marker in assays. The fluorescence was examined by confocal microscopy (Olympus FV1200).

## Gene expression assays in guard cells
The full length promoters (2000 bp) of four genes (*GhCBL1A1*, *GhCBL2A1*, *GhCIPK6D1* and *GhCIPK6D3*) were cloned without mutation into the pKGBFS7 vector by Gateway technology. The resulting plasmid was transferred to *Agrobacterium tumefaciens* strain GV3101 for inoculation of tobacco leaves. The process was same as that described above (subcellular localization and BiFC assays). GFP fluorescence was detected under a confocal microscope (Olympus FV1200). For GUS histochemistry the inoculated tobacco leaves were incubated in GUS dye solution (Coolaber, SL7160) at 25–37 °C for 1 h or overnight. The leaves were sequentially decolorized with 70%, 80% and 85% ethanol over several hours until the material bleaches white. Stomatal imaging was performed using light microscopy (Zeiss Axio Scop A1, Oberkochen, Germany). The primers used are listed in Supplementary Data 4.

Isolation of guard-cell enriched tissue for RNA extraction assay was carried out as described by Jalakas et al.[71]. In brief, leaves were mixed and broken in ice water, and filtered through nylon mesh to obtain guard cell rich fragments; RNA was extracted after the guard cells were obtained. Specific primers were designed for 4 genes and qRT-PCR experiments were performed, using *GhUB7* as an internal control. RNA extraction and qRT-PCR assays were conducted following the same procedures described above (*CIPK6* gene expression analysis).

## In vitro pull-down assay
For in vitro pull-down assays, the full-length CDS sequences of *GhCBL1A1* and *GhCIPK6D1* were cloned into vectors PET-28-a (Novagen) and pGEX-4T-1 (Pharmacia) respectively. The vector constructs GhCBL1A1-His and GhCIPK6D1-GST were transformed into *E. coli* strain *BL21*. Empty GST and recombinant GhCIPK6D1-GST protein were used to pull down GhCBL1A1 protein. Protein samples were purified using the MagneGST Protein Purification System (Cat. #V8600, Promega) and the MagneHis Protein Purification System (Promega) according to the manufacturer's instructions. The pull-down assay was performed similarly to the previous study[72]. In brief, the various protein combinations were placed on a silent mixer at 4 °C for 1 h. After that, GSTSep Glutathione Agarose Resin was added, and the mixture was left to incubate for 2–3 h. Then, the proteins were washed using a washing buffer (20 mM Tris, 150 mM NaCl) and subjected to Western blotting analysis using anti-GST (1:5000; ABclonal) and anti-His (1:5000; ABclonal) antibodies. The primers used are listed in Supplementary Data 4.

## Statistical analysis
Statistical significance was assessed by one-way ANOVA followed by Duncan's multiple range test or two-tailed Student's *t*-test (see each figure legend). All analyses were performed by SPSS (SPSS 20.0; SPSS, Chicago, IL, USA).

## Reporting summary
Further information on research design is available in the Nature Portfolio Reporting Summary linked to this article.

## Data availability
All data generated from this study are available in the article and Supplementary Information files. All constructs and transgenic plants are available upon request. Gene, CDS and promoter sequences of the *CIPK6* genes are provided in Supplementary Data 2. Source data are provided with this paper.

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

## Acknowledgements

This work was supported by funding from National Natural Science Foundation of China, no. 32171942 (XY.Y.), and 32301754 (WN.S.), the National Key Project of Research and the Development Plan of China, 2021YFF1000103 (XY.Y.), and Bintuan Science and Technology Program, no. 2022DB012 (XY.Y.).

## Author contributions

XY.Y. and XL.Z. designed and supervised the project. WN.S. and LJ.X. performed experiments and wrote the draft manuscript. SM.S., JW.D. and DD.Y. helped perform phenotypic assays. JQ.Y. and MJ.W. helped in analyzing gene evolution. SX.J., LF.Z. and K.L. organized the results and revised the manuscript.

## Competing interests

The authors declare no competing interests.
