## [Peer Review File · Nature Communications]

Evolution and subfunctionalization of CIPK6 homologous genes in regulating drought resistance in cottonEditorial Note: Parts of this Peer Review File have been redacted as indicated to remove third-party material where no permission to publish could be obtained. When text is deleted in rebuttals and referee reports, add “[redacted]” in that location.

REVIEWER COMMENTS

Reviewer #1 (Remarks to the Author):

The manuscript by Sun et al., investigates the evolutionary relationship of eight GhCIPK6 homologous genes in upland cotton, and found that GhCIPK6D1 negatively regulated cotton drought resistance, while GhCIPK6D3 was a positive regulator, indicating that functional differentiation was obvious during the evolution of the gene family. The negative/positive synergistic regulation of GhCBL1A1-GhCIPK6D1 and GhCBL2A1-GhCIPK6D3 on drought resistance of cotton was further confirmed by phenotypic analysis, biochemical interaction, bioinformatics analysis and genetic transformation. The authors could improve and clarify of their presentation and interpretation of some conclusions. There are some comments that the authors should address to solidify their findings and enhance the overall quality of the presentation.

1. I am very doubt that the authors used Non-invasive Micro-test Technology (NMT) to measure the velocity of K⁺ flux in the guard cells of cotton. The detailed methods should be provided. Is there any references? How many guard cells were measured?
2. In the lines 178-184, the drought stress phenotypes of ko#5 and ko#6 appeared normal compared to WT, but there were significant differences in the determination of MDA and H₂O₂. The authors need to explain the reason. Besides it is recommended to add some more intuitive physiological data to make the results more clear, such as leaf temperature, because the differences in phenotype are not very obvious. In addition, the author needs to explain why MDA and H₂O₂ are measured, and cite the corresponding references.
3. In the stomatal phenotype and related data statistics of Fig.3, if the same type of WT is used, why are there such big differences in WT under the same treatment conditions? And why under normal conditions, GhCIPK6D3 transgenic lines and RNAi lines have significant differences in stomatal aperture and K⁺ flux, but there is no significant difference with WT in the phenotype of Fig.2, and the author also did not describe the growth status of GhCIPK6D3 transgenic lines and RNAi lines under normal conditions in the manuscript. The author needs to explain the reasons for the differences.
4. In Fig.3f, according to the statistics of stomatal opening results in Fig.3d, there is almost no difference in stomatal opening of OE24/OE35 under drought conditions, but the flux of K⁺ is quite different. Similarly, there is a difference in stomatal opening of Ri16/Ri19 but almost no difference in K⁺ flux. The authors need to explain the reason.
5. 1) When transitioning from CIPK6 to candidate CBLs, it is recommended to explain the reasons for selecting CBL1 and CBL2 rather than other CBLs. 2) In the Fig.4 Y2H experiment, the specific information of the positive control and the negative control needs to be supplemented. 3) The marker CBL1n used in Fig.4d was not clearly described in the figure legends and the manuscript, and no relevant literatures were cited, and no specific statistical data analysis was performed. 4) It is recommended to add co-located plasma membrane markers in Fig.4e. In addition, obvious nuclear localization can also be observed from the figure, and the author needs to explain it. 5) In Fig.4f, it is recommended to mark the size of the protein band clearly. In addition, which band of the GST lane of IP is the target protein or what? The results are not convincing and need to repeat. 6) The interaction was confirmed in vivo by heterologous tobacco, which is recommended to be repeated in cotton. 7) The evidence of interaction between GhCBL2A1 and GhCIPK6D3 and the results of subcellular localization are insufficient, and relevant literatures should be cited if there are existing reports. Minor comments:

1. The description in line 192 “with Ri19 showing lower WUE compared to WT” are one-sided, and there are also differences in Ri16, so the results need to be re-described accurately.
2. There are many mistakes that should not be made in the figures and figure legends. The authors should take them more seriously and not let the reviewers doubt their professionalism. 1) In Fig.2, the descriptions of a-b and c-d in the figure legends are inconsistent with the picture. 2) In Fig.5, “f” and “g” are mislabeled. 3) The “e” in Supplementary Fig.6 legends should be “d”. 4) In Supplementary

Fig.10, the annotations a, b, and c are missing. 5) In Supplementary Fig.11, the d and e are labeled incorrectly. 6) In Supplementary Fig.13, the annotations a, b, and c are missing.

3. Line 311 should be "As GhCIPK6D3 could interact with GhCBL2A1".

4. In Fig.8, the GhCBL1A1-GhCIPK6D1 module is preferably anchored on the plasma membrane rather than distributed in the cytoplasm.
5. The Discussion section repeats much of the content of the Results section and requires to be carefully revised to make a concise summary of the findings and their significance in the context of the existing literature. At the same time, Discussion should explain the model shown in Figure 8 more clearly, rather than a simple summary.
6. Fig.4f has no tagged protein size.

Reviewer #2 (Remarks to the Author):

Sun et al., presented a study regarding the functional divergence of duplicated CIPK6 genes in cotton. The authors compared the CIPK6 gene numbers, expression response, and interaction patterns in upland cotton. They found the GhCIPK6D1 and GhCIPK6D3 exhibited different functions in response to drought stress treatment, which indicates functional differentiation after the polyploidization event in cotton. Further genetic and biochemistry experiments confirmed the synergistic negative/positive regulation of these two genes for drought resistance. They also found the differential interaction partner (CBL) which could regulate stomatal movement by control the flow direction of K⁺ in guard cells.

#Major concerns#

The evolutionary analysis of CIPK6 gene family needs much improvement. It is not clear when and how (by what mechanism) these four copies of CIPK6 evolved. Although the fig 1a showed 23 species and the numbers of CIPK6 in each species, it is rather confusing about relatively similar number of CIPK6 genes in these species given different WGD history. How the CIPK6 originated and retained after these marked WGDs?

The current version mainly analyzed the GhCIPK6s, and only expression patterns were shown for their diploid ancestor. It might need parallel experiments for the functions of these four CIPK6s in both *G. arboreum* and *G. raimondii*, and we can then discuss about the sub-functionalization.

Another concern is about the GhCIPK6D3, which only have RNAi lines. But the authors didn't present the known down level, and I am also not sure whether it could also affect the other CIPKs. Further verification of the expression changes of all CIPKs should be added.

#Minors#

Phylogenetic trees (Fig S1b, Fig 1b) need to show bootstrap values for confidently understanding the evolutionary history of the gene families.

More details are needed for the comparison of CIPK6 genes under drought stress. For example, what's the development stage of the treated plants? Whether the diploid and tetraploid are in the same stage?

Line 161-170, it's quite surprising the promoter regions of the CIPK6 genes exhibiting such large divergence given only 1MA divergence.

The phenotype of stomatal aperture of CIPK6D3 OE and RNAi lines seems no much difference under normal or drought stress treatments. I can't see any statistical significance test there.

Reviewer #3 (Remarks to the Author):

The CBL-CIPK modules in abiotic stress responses have been extensively studied in plants. In the manuscript titled "Evolution and subfunctionalization of CIPK6 homologous genes in regulating drought resistance in cotton", Sun et al. reported that both CIPK6D1 and CIPK6D3 in *Gossypium hirsutum* were transcriptionally induced by drought. Interestingly, overexpressing GhCIPK6D1 or GhCIPK6D3 resulted in an opposite effect on stomatal movement. The authors further reported that CIPKs in cotton are sequentially evolved, with an order of 6A1/D1 to 6A4/D4 to 6A3/D3/6A2/D2. They proposed that CIPK6D3 was originated from CIPK6D1 but became functionally opposite to CIPK6D1 to better adapt to environment changes.

Although the study provided a novel view on functionality of CIPK, it does not contain mechanistic insights and sufficient supporting data to justify publication in its current form.

Major issues:

1. The claim that GhCIPKs function in stomatal movement needs to be supported by the expression of these genes, i.e. GhCIPK6D1, GhCIPK6D3, GhCBL1A1, GhCBL2A1, in guard cells.
2. To support stomatal movement and drought tolerance, stomatal aperture as shown in Fig 2 is not sufficient. Parameters, such as leaf water loss and leaf temperature, are needed.
3. Because stomatal density and guard cell size also have an impact on water use efficiency, they should be examined to exclude the possibility that stomatal aperture is not the sole/major factor.
4. Line 226, the rationale that only CBL1/2 were chosen as candidate interacting proteins of GhCIPK6 for drought tolerance is not clear to me. Why were other CBLs excluded?
5. In Fig 4-d,4-e, both GhCIPK6D1 and GhCBL1A1 were detected in the nucleus and interacting between these two occurred in the nucleus, in addition to the PM. Were these signals real? The authors should explain its possible biological relevance.
6. In Fig 4-c, positive interactions between GhCBL1A1 and GhCIPK6D1, GhCIPK6D3 were detected. So what's the rationale to exclude the biological relevance of the GhCBL1A1 and GhCIPK6D3 complex?
7. In Fig 8, tonoplast localization of GhCIPK6D3 and GhCBL2A1 should be demonstrated to support their function on vacuolar membranes.

Response to reviewers

We are grateful to you and the reviewers for the comments and suggestions on our manuscript. We have addressed each of the points raised, including carrying out additional work - our point-by-point responses are in red, below. We have uploaded a version of the manuscript that highlights all textual changes that address the reviewers' comments, and a clean version.

We hope the manuscript is suitable now for publication - we feel the suggestions have strongly improved the manuscript.

With best wishes

Prof. Xiyan Yang, on behalf of the authors.

Reviewer comments:

Reviewer #1 (Remarks to the Author):

The manuscript by Sun et al., investigates the evolutionary relationship of eight GhCIPK6 homologous genes in upland cotton, and found that GhCIPK6D1 negatively regulated cotton drought resistance, while GhCIPK6D3 was a positive regulator, indicating that functional differentiation was obvious during the evolution of the gene family. The negative/positive synergistic regulation of GhCBL1A1-GhCIPK6D1 and GhCBL2A1-GhCIPK6D3 on drought resistance of cotton was further confirmed by phenotypic analysis, biochemical interaction, bioinformatics analysis and genetic transformation. The authors could improve and clarify their presentation and interpretation of some conclusions. There are some comments that the authors should address to solidify their findings and enhance the overall quality of the presentation.

1. I am very doubt that the authors used Non-invasive Micro-test Technology (NMT) to measure the velocity of K⁺ flux in the guard cells of cotton. The detailed methods should be provided. Is there any references? How many guard cells were measured? **Reply:** Thanks for your comments. Non-invasive Micro-test Technology (NMT) is used by many research institutes, and the relevant research have been published in many top journals. We added the details of the K⁺ flux measurement method by NMT and cited a reference (Line 607-608), Phosphorylation of the plasma membrane H⁺-ATPase AHA2 by BAK1 is required for ABA-induced stomatal closure in Arabidopsis. *Plant Cell*, 2022). The assay for the measurement of K⁺ flux, three biological replicates were measured, and three guard cells were randomly selected for each biological replicate.

2. In the lines 178-184, the drought stress phenotypes of ko#5 and ko#6 appeared normal compared to WT, but there were significant differences in the determination of MDA and H₂O₂. The authors need to explain the reason. Besides it is recommended to add some more intuitive physiological data to make the results clearer, such as leaf temperature, because the differences in phenotype are not very obvious. In addition, the author needs to explain why MDA and H₂O₂ are measured, and cite the corresponding references.

Reply: Thank you for your comments. We were sorry that we made an unclear statement here. In Fig. 2a, the growth of all the plants was equal before drought treatment. However, after drought treatment, the OE lines (OE4, OE12) of *GhCIPK6D1* showed drought sensitive phenotype with many true leaves wilting; mutant lines (*ko#5*,

ko#6) showed resistance phenotype compared to WT with only cotyledons wilting (we use arrows to indicate this in the image below). And changed accordingly in the revised manuscript (Line 182-185).

Fig. 2a

The phenotypes were also consistent with the differences in MDA and H₂O₂ content among different materials after drought treatment. MDA and H₂O₂ can better characterize the damage degree of plant cells after drought stress. Published studies have also used MDA and H₂O₂ to characterize drought correlation (i.e. Li et al., N6-methyladenosine RNA modification regulates cotton drought response in a Ca²⁺ and ABA-dependent manner. *Plant Biotechnol J.* 2023). In this assay, we also measured other drought-related indicators, for example, transpiration rate, relative water content, proline content and water use efficiency in the revised Fig. 2 and supplementary Fig. 8. These results were consistent with the phenotype of the plants, which well proved that *GhCIPK6D1* negatively regulated cotton drought resistance, and *GhCIPK6D3* positively regulated cotton drought resistance.

Fig. 2c and 2f

supplementary Fig. 7

3. In the stomatal phenotype and related data statistics of Fig.3, if the same type of WT is used, why are there such big differences in WT under the same treatment conditions? And why under normal conditions, *GhCIPK6D3* transgenic lines and RNAi lines have significant differences in stomatal aperture and K⁺ flux, but there is no significant difference with WT in the phenotype of Fig. 2, and the author also did not describe the growth status of *GhCIPK6D3* transgenic lines and RNAi lines under

normal conditions in the manuscript. The author needs to explain the reasons for the differences.

Reply: Thanks for your comments. The transgenic backgrounds of *GhCIPK6D1* (Jin668 as transgenic receptor) and *GhCIPK6D3* (YZ1 as transgenic receptor) are different. The two cotton germplasms were widely used for cotton transformation. As a result, there might exist variations in drought resistance phenotypes between the two genotypes. Consequently, there could also be discrepancies observed in stomatal phenotypes when subjected to the drought treatment.

As you said, we also observed that *GhCIPK6D3* could affect some index under normal condition, such as K^+ transport and stomatal movement. It is possible that *GhCIPK6D3* might be involved in additional functions, which could influence cotton development to a certain extent. However, under conditions of adequate water supply, the impact of *GhCIPK6D3* on functions such as stomata may not be sufficient to cause delayed cotton development, therefore showing no significant difference in phenotype compared to WT. In our manuscript, we provided a description of the growth status of the *GhCIPK6D3* transgenic lines and the RNAi lines under normal conditions (Line 197-198).

4. In Fig.3f, according to the statistics of stomatal opening results in Fig.3d, there is almost no difference in stomatal opening of OE24/OE35 under drought conditions, but the flux of K^+ is quite different. Similarly, there is a difference in stomatal opening of Ri16/Ri19 but almost no difference in K^+ flux. The authors need to explain the reason.

Reply: Thanks for your comments. The results of stomatal phenotype and K^+ flux rate were the repeated results we obtained from multiple biological replicates. From the data, we could not observe much different in stomatal opening of OE24/OE35 under drought conditions, while the flux of K^+ is quite different. It might be because that the data scale of K^+ flux is much higher than that of stomatal opening, especially at high level of K^+ flux. And the mechanism of stomatal regulation is relatively complex. Since we have found some targets related to K^+ channels, we focus on the regulation of K^+ on stomata. Besides K^+ , other factors including ABA signaling and proton transport may also affect stomatal movement. Therefore, changes in K^+ flux may not have a linear relationship with changes in stomatal movement. Overall, we believe that the K^+ flux of guard cell and stomatal movement were partially according with each other in *GhCIPK6D3* transgenic plants.

5. 1) When transitioning from CIPK6 to candidate CBLs, it is recommended to explain the reasons for selecting CBL1 and CBL2 rather than other CBLs.

Reply: We cloned some representative gene members (homologous genes from Arabidopsis) of *GhCBL* subfamilies for point-to-point interaction analysis in upland cotton, and found that *GhCIPK6D1* only interact with members of *GhCBL1* subfamilies (*GhCBL1A1* and *GhCBL9A1*, see the image below), and *GhCIPK6D3* interact with members of *GhCBL1* (*GhCBL1A1/1D1* and *GhCBL9A1*) and *GhCBL2* (*GhCBL2A1*) subfamilies (this has been reported in our previous research, Deng et al., The calcium sensor CBL2 and its interacting kinase CIPK6 are involved in plant sugar homeostasis via interacting with tonoplast sugar transporter TST2. *Plant Physiology*, 2020). So, we selected CBL1 and CBL2 subfamily genes for interaction analysis between CIPK6s and CBL1/2s in *Gossypium*.

[Redacted]
Deng et al., 2020 *Plant Physiol*

2) In the Fig. 4 Y2H experiment, the specific information of the positive control and the negative control needs to be supplemented.

Reply: In the Y2H experiment, “Po” represents positive control and “Ne” represents negative control. The positive control was the diploid hybrid yeast containing pGBKT7-53 and pGADT7-T, the negative control was the diploid hybrid yeast containing pGBKT7-Lam and pGADT7-T. We also added those to the legend of Fig. 4.

3) The marker CBL1n used in Fig. 4d was not clearly described in the figure legends and the manuscript, and no relevant literatures were cited, and no specific statistical data analysis was performed.

4) It is recommended to add co-located plasma membrane markers in Fig. 4e. In addition, obvious nuclear localization can also be observed from the figure, and the author needs to explain it.

6) The interaction was confirmed in vivo by heterologous tobacco, which is recommended to be repeated in cotton.

Reply: Thank you for your suggestion.

The marker CBLn (CBLn:RFP) is a plasma membrane localized marker widely used. We added a literature in the revised manuscript (Batistic et al., Dual fatty acyl modification determines the localization and plasma membrane targeting of CBL/CIPK Ca²⁺ signaling complexes in Arabidopsis. *Plant Cell*, 2008). And we added the description in the figure legend.

We re-validated the localization (Fig. 4e) and interaction between GhCBL1A1 and GhCIPK6D1 in cotton protoplasts (Fig. 4g), added the plasma membrane marker (CBLn:RFP) and nucleus marker (HY5:RFP), and cited the literature for nucleus marker in revised manuscript (Lin et al., B-BOX DOMAIN PROTEIN28 negatively regulates photomorphogenesis by repressing the activity of transcription factor HY5 and undergoes COP1-mediated degradation. *Plant Cell*, 2018).

The results showed that both GhCBL1A1 and GhCIPK6D1 were located to the plasma membrane and nucleus (Fig. 4d, e), while displayed high interaction signal on the plasma membrane other than on the nucleus (Fig. 4f, g).

5) In Fig.4f, it is recommended to mark the size of the protein band clearly. In addition, which band of the GST lane of IP is the target protein or what? The results are not convincing and need to repeat.

Reply: Thank you for your suggestion and comment. We repeated the pull-down assay between GhCBL1A1 (His-GhCBL1A1) and GhCIPK6D1 (GST-GhCIPK6D1) (Fig. 4h, as shown in the following figure), and we also added the size of marker and the candidate protein. It could be found that the His-GhCBL1A1 could be pull down by GST-GhCIPK6D1, which confirm the interaction between GhCBL1A1 and GhCIPK6D1.

7) The evidence of interaction between GhCBL2A1 and GhCIPK6D3 and the results of subcellular localization are insufficient, and relevant literatures should be cited if there are existing reports.

Reply: The subcellular localization of GhCBL2A1 (formerly named GhCBL2) and GhCIPK6D3 (formerly named GhCIPK6), and the interaction between GhCBL2A1 and GhCIPK6D3 was reported in our previous study (Deng et al., The calcium sensor CBL2 and its interacting kinase CIPK6 are involved in plant sugar homeostasis via interacting with tonoplast sugar transporter TST2. *Plant Physiology*, 2020). It was reported that GhCBL2A1 was located to the vacuolar membrane, and GhCIPK6D3 was located to the plasma membrane and nucleus, BiFC and LUC experiments proved that they interact on the vacuole membrane (red arrows indication). These results were also cited in the revised manuscript.

[Redacted]
Deng et al., 2020 *Plant Physiol*

Minor comments:

1. The description in line 192 “with Ri19 showing lower WUE compared to WT” are one-sided, and there are also differences in Ri16, so the results need to be re-described accurately.

Reply: Thank you for your comment. We are sorry about the incomplete description. We have re-described the results of the WUE accurately in the article (Line 202-204). And we have replaced the results of MDA and Tr in Fig. 2, and the results of WUE in supplementary Fig. 7d to unify the detection indicators of *GhCIPK6D1* and *GhCIPK6D3* transgenic plants.

2. There are many mistakes that should not be made in the figures and figure legends. The authors should take them more seriously and not let the reviewers doubt their professionalism. 1) In Fig.2, the descriptions of a-b and c-d in the figure legends are inconsistent with the picture. 2) In Fig.5, “f” and “g” are mislabeled. 3) The “e” in Supplementary Fig.6 legends should be “d”. 4) In Supplementary Fig.10, the annotations a, b, and c are missing. 5) In Supplementary Fig.11, the d and e are labeled incorrectly. 6) In Supplementary Fig.13, the annotations a, b, and c are missing.

Reply: Thank you for pointing out the mistakes, we have checked the numbers in all figures and the corresponding descriptions of legends, and corrected the mistakes. And we have checked the whole manuscript carefully to change any mistakes.

3. Line 311 should be “As GhCIPK6D3 could interact with GhCBL2A1”. **Reply:** Thank you for pointing out the mistakes, we have corrected it. (Line 336-337).

4. In Fig.8, the GhCBL1A1-GhCIPK6D1 module is preferably anchored on the plasma membrane rather than distributed in the cytoplasm.

Reply: Thank you for your comment, we have redrawn the module and made corrections (the red box indicates as shown in below).

5. The Discussion section repeats much of the content of the Results section and requires to be carefully revised to make a concise summary of the findings and their significance in the context of the existing literature. At the same time, Discussion should explain the model shown in Figure 8 more clearly, rather than a simple summary. **Reply:** Thank you for your suggestions, we have revised the discussion section, and described the model in detail and clearly in the revised manuscript.

6. Fig.4f has no tagged protein size.

Reply: We have added proteins size on the repeated pull-down assay result (as shown the comment of point 5 above).

Reviewer #2 (Remarks to the Author):

Sun et al., presented a study regarding the functional divergence of duplicated CIPK6 genes in cotton. The authors compared the CIPK6 gene numbers, expression response, and interaction patterns in upland cotton. They found the GhCIPK6D1 and GhCIPK6D3 exhibited different functions in response to drought stress treatment, which indicates functional differentiation after the polyploidization event in cotton. Further genetic and biochemistry experiments confirmed the synergistic negative/positive regulation of these two genes for drought resistance. They also found the differential interaction partner (CBL) which could regulate stomatal movement by control the flow direction of K⁺ in guard cells.

#Major concerns#

1. The evolutionary analysis if CIPK6 gene family needs much improvement. It is not clear when and how (by what mechanism) these four copies of CIPK6 evolved. Although the fig 1a showed 23 species and the numbers of CIPK6 in each species, it is rather confusing about relatively similar number of CIPK6 genes in these species given different WGD history. How the CIPK6 originated and retained after these marked WGDs?

Reply: Thank you for your suggestions. In order to fully analyze the evolution of *CIPK6* gene family, we have made more improvements. 1) We increased the number of

species analyzed, adding some moss and fern species to determine the period in which *CIPK6* appeared. The results were consistent with what we have done in the original manuscript, the *CIPK6* gene appeared during the differentiation of monocotyledonous and dicotyledonous plants. 2) Through two rounds of genome replication, there should have been at least six genes in *G. arboretum* and *G. raimonddi*, we further analyzed gene duplication and evolution in polyploidization-diploidization cycles in plants. The results indicated that the main mechanism for replication is WGD in *G. arboretum* and *G. raimonddi*, there was no tandem duplication (TD), proximal duplication (PD), transposed duplication (TRD), dispersed duplication (DSD) (supplementary Fig. 2b), as shown in below. 3) The evolutionary order of the four genes was 6A1/D1-6A4/D4-6A2/D2 and 6A3/D3, which can be proved by the evolutionary analysis of diploid *Gossypium*, matched with related *Malvaceae* species, and other tetraploid *Gossypium* (supplementary Fig. 1b). 4) Finally, eight genes of *GhCIPK6* were formed in upland cotton by cross quadrupling of two diploid cotton (supplementary Fig. 1a).

2. The current version mainly analyzed the GhCIPK6s, and only expression patterns were shown for their diploid ancestor. It might need parallel experiments for the functions of these four CIPK6s in both *G. arboretum* and *G. raimonddi*, and we can then discuss about the sub-functionalization.

Reply: Thank you for your comment. Firstly, *G. arboretum* and *G. raimonddi* are perennial diploid wild cotton, their diploid genome is the ancestor of the widely cultivated tetraploid cotton (*G. hirsutum*), they are the donors of the At and Dt subgenomes, respectively. Secondly, there are significant differences between the two-diploid ancestor. Especially *G. raimonddi*, the annual seed harvest is very low, it is very difficult to harvest enough seeds. Thirdly, we pay more attention to the differences in the evolution of *Gossypium*, through expression patterns, we found that the expression of *CIPK6* and *CBL1/2* family genes were different between diploid ancestors and upland cotton, indicating that there were functional differences in upland cotton after polyploidization. So, we further studied the sub-functionalization and its regulatory mechanism of *CIPK6s* in upland cotton.

3. Another concern is about the GhCIPK6D3, which only have RNAi lines. But the authors didn't present the known down level, and I am also not sure whether it could also affect the other CIPKs. Further verification of the expression changes of all CIPKs should be added.

Reply: Thanks for your suggestion. The mRNA levels of all *CIPK6s* in the RNAi lines (Supplementary Fig. 8) was shown in the following figure. The results showed that Ri16 and Ri19 lines down-regulated the expression of *GhCIPK6A1/D1/A3/D3/A4/D4*, of which 6A1/D1/A4/D4 may not affect the drought resistance of RNAi lines due to its low expression, although no expression was detected in some RNAi lines. Ri16

increased the expression of *GhCIPK6A2/D2*, but Ri19 showed the opposite trend. Therefore, the relative drought sensitivity of RNAi lines may be mainly due to the reduced expression level of highly expressed *GhCIPK6A3/D3*.

#Minors#

1. Phylogenetic trees (Fig S1b, Fig 1b) need to show bootstrap values for confidently understanding the evolutionary history of the gene families.

Reply: Thank you for your suggestions, we have modified and refined the phylogenetic trees (as shown in the following figures).

2. More details are needed for the comparison of CIPK6 genes under drought stress.

For example, what's the development stage of the treated plants? Whether the diploid and tetraploid are in the same stage?

Reply: For drought-induced expression analysis of *CIPK6* genes, the cotton seeds of *G. arboreum* and *G. hirsutum* were planted in the culture room (25°C with 16 h light/8 h dark photoperiod, and grown in nutrient soil), and well-watered until three-leaves stage. Then the plants were subjected to drought treatment by withdrawing water for 14 days, and the normal watered plants were used as parallel control. Samples were collected at 10 days after withdrawing water. The *G. raimondii* is a perennial wild cotton. It is difficult to obtain the seed from plants. We obtained the samples from greenhouse used parallel drought condition. We added those in the material methods section of the revised manuscript.

3. Line 161-170, it's quite surprising the promoter regions of the *CIPK6* genes exhibiting such large divergence given only 1MA divergence.

Reply: Thank you for your suggestions. The variations among those promoter regions of the *CIPK6* genes were appeared in 1MA, they were originated from the two round WGD for very long time. For the promoter analysis, the 2000bp upstream sequence of each *CIPK6* gene was selected. It could be observed that there are some differences in the type and number of promoter elements among *CIPK6* genes, with relatively large differences in the G-box and Box4 elements, which may be due to the evolution process. But there were less variations from diploid (*G. arboreum* or *G. raimondii*) to tetraploid (*G. hirsutum*). We remapped the order of the genes (Supplementary Fig. 4).

1	0	0	0	1	0	0	2	1	0	4	1	0	0	0	0	0	0	0	0	1	0	1	0	0	0	0	0	0	1	6	0	0	0	6	0	1	0	0	0	0	0	0	0	0	1	GrCIPK6D1			
1	0	0	0	0	0	0	1	1	0	2	1	0	0	0	0	0	0	0	0	1	0	2	0	0	0	0	0	0	1	3	0	0	1	2	1	2	0	1	0	0	0	1	1	0	2	GhCIPK6D1			
1	0	0	1	0	0	1	0	0	0	2	1	0	2	2	0	1	0	0	0	0	0	0	0	0	0	0	0	1	1	4	1	0	0	6	0	1	0	4	2	0	0	0	1	0	2	GaCIPK6A1			
1	0	0	1	0	0	1	0	0	0	2	1	0	1	1	0	1	0	0	0	0	0	0	0	0	0	0	0	1	0	3	1	0	0	5	0	1	0	4	0	0	0	0	0	0	2	GhCIPK6A1			
0	0	2	0	0	2	0	2	1	0	6	0	0	0	0	0	0	0	0	0	0	1	0	0	0	0	0	0	0	8	0	0	0	5	0	0	0	0	0	0	1	0	2	0	0	2	GrCIPK6D4			
0	0	2	0	0	2	0	1	1	0	8	0	0	1	1	0	0	1	1	0	0	0	1	1	0	0	0	0	0	11	0	0	0	6	0	0	0	0	0	1	0	1	0	0	2	GhCIPK6D4				
0	0	0	0	0	2	1	0	1	0	2	1	0	1	1	0	0	1	3	0	1	0	1	0	0	0	0	2	2	2	3	0	0	6	0	0	0	0	0	0	2	0	1	0	0	1	GaCIPK6A4			
0	0	2	0	0	2	1	1	1	0	3	1	0	1	1	0	0	0	1	0	1	0	1	0	0	0	1	4	3	0	0	5	0	0	0	0	2	0	1	0	1	0	0	0	1	GhCIPK6A4				
1	0	0	0	0	0	0	0	0	1	0	0	0	0	0	0	2	0	0	1	1	1	0	0	1	0	1	0	0	0	0	0	5	0	1	0	3	0	0	1	0	0	0	0	0	0	GrCIPK6D3			
0	0	0	0	0	0	0	0	1	6	1	0	3	3	1	3	1	0	0	0	0	0	0	0	0	0	0	1	9	2	0	0	5	0	0	0	0	0	0	0	0	0	2	2	0	0	0	1	GhCIPK6D3	
0	1	0	0	0	0	0	0	0	1	7	0	1	1	1	1	1	0	0	0	0	0	0	0	0	0	0	1	9	1	0	0	5	0	0	0	0	0	0	0	0	0	0	1	0	0	0	GaCIPK6A3		
0	1	0	0	0	0	0	0	0	1	10	1	0	2	2	2	1	2	0	0	1	0	0	0	0	0	1	13	0	0	0	4	0	0	0	0	0	0	0	0	0	0	0	0	0	0	1	GhCIPK6A3		
0	0	0	1	1	0	0	1	1	1	2	0	0	1	1	2	0	0	3	2	0	0	1	0	0	0	0	0	0	0	0	0	0	2	0	0	2	0	0	0	0	0	1	1	0	1	0	1	GrCIPK6D2	
1	0	0	1	0	0	0	0	1	0	3	0	0	1	1	0	1	0	2	0	0	0	0	0	0	1	0	4	2	0	1	6	0	2	0	3	0	0	0	0	1	0	0	0	0	0	0	GhCIPK6D2		
0	0	0	1	1	0	0	0	0	1	0	0	0	0	0	0	0	0	0	0	0	0	0	0	0	0	0	0	0	0	0	0	0	2	1	2	0	0	0	1	0	0	1	0	0	1	GaCIPK6A2			
1	0	0	0	0	0	0	0	0	0	3	0	0	1	1	0	1	0	0	0	0	0	0	0	0	1	0	4	1	0	0	10	0	1	0	3	0	1	0	0	0	1	0	0	0	1	0	0	1	GhCIPK6A2

Plant growth and development

Plant hormones responsive

Abiotic stress responsive

Light responsive

4. The phenotype of stomatal aperture of CIPKD3 OE and RNAi lines seems no much difference under normal or drought stress treatments. I can't see any statistical significance test there.

Reply: Thanks for your comments. The stomatal aperture reaches the micron level, so it could not observe the differences from those pictures. However, we did statistical significance analysis for those data using one-way ANOVAs followed by Duncan's multiple range test, and the significance was indicated as a-e for normal condition and f-i for drought condition in Fig. 3d.

Reviewer #3 (Remarks to the Author):

The CBL-CIPK modules in abiotic stress responses have been extensively studied in plants. In the manuscript titled “Evolution and subfunctionalization of CIPK6 homologous genes in regulating drought resistance in cotton”, Sun et al. reported that both CIPK6D1 and CIPK6D3 in *Gossypium hirsutum* were transcriptionally induced by drought. Interestingly, overexpressing GhCIPK6D1 or GhCIPK6D3 resulted in an opposite effect on stomatal movement. The authors further reported that CIPKs in cotton are sequentially evolved, with an order of 6A1/D1 to 6A4/D4 to 6A3/D3/6A2/D2. They proposed that CIPK6D3 was originated from CIPK6D1 but became functionally opposite to CIPK6D1 to better adapt to environment changes.

Although the study provided a novel view on functionality of CIPK, it does not contain mechanistic insights and sufficient supporting data to justify publication in its current form.

Major issues:

1. The claim that GhCIPKs function in stomatal movement needs to be supported by the expression of these genes, i.e. GhCIPK6D1, GhCIPK6D3, GhCBL1A1, GhCBL2A1, in guard cells.

Reply: Thanks for your suggestion. In order to better demonstrate the expression of the four genes (*GhCBL1A1*, *GhCBL2A1*, *GhCIPK6D1* and *GhCIPK6D3*) in guard cells, three assays (gene specific promoter::GFP, gene specific promoter::GUS and qRT-PCR test for guard cell) were performed. 1) The promoter of the four genes were fused to GFP-GUS protein expression vector and transformed into tobacco leaves, the results showed that the fluorescence signals of GFP protein were observed in the guard cells for each gene (supplementary Fig.18a); 2) The GUS signals were also found in guard cells for each gene (supplementary Fig.18b). 3) The guard cells of cotton leaves were isolated, and the RNA were extracted for the qRT-PCR analysis of the four genes in the guard cells, using the expression of the whole leaves as control. The results showed that the four genes were preferential expression in the guard cells, accounting for 74.75%, 62.11%, 66.96% and 64.13% of the total expression of leaves, respectively (supplementary Fig. 18c). Those results and relative methods were added in the revised manuscript.

2. To support stomatal movement and drought tolerance, stomatal aperture as shown in

Fig 2 is not sufficient. Parameters, such as leaf water loss and leaf temperature, are needed.

Reply: Thanks for your suggestion. We also obtained other drought-related indicators, for example, transpiration rate (Tr), relative water content, and water use efficiency related to stomatal movement (as shown in the revised Fig. 2 and supplementary Fig. 7c, 7d).

Fig. 2c and 2f

supplementary Fig. 7c, 7d

3. Because stomatal density and guard cell size also have an impact on water use efficiency, they should be examined to exclude the possibility that stomatal aperture is not the sole/major factor.

Reply: Thank you for your suggestions. We also measured the stomatal density for different plants and added the results and figures in the revised manuscript (supplementary Fig. 9, 15c and 17). The results showed that GhCBL1A1-GhCIPK6D1 did not affect stomatal density, but overexpression of GhCBL2A1-GhCIPK6D3 decreased stomatal density, and its drought resistance might be related to the change of stomatal density to some extent.

supplementary Fig. 9

supplementary Fig. 15c

Normal Drought supplementary Fig.

17

4. Line 226, the rationale that only CBL1/2 were chosen as candidate interacting proteins of GhCIPK6 for drought tolerance is not clear to me. Why were other CBLs excluded?

Reply: We cloned some representative gene members (homologous genes from Arabidopsis) of *GhCBL* subfamilies for point-to-point interaction analysis in upland cotton, and found that GhCIPK6D1 only interact with members of GhCBL1 subfamilies (GhCBL1A1 and GhCBL9A1, see the image below), and GhCIPK6D3 interact with members of GhCBL1 (GhCBL1A1/1D1 and GhCBL9A1) and GhCBL2 (GhCBL2A1) subfamilies (this has been reported in our previous research, Deng et al., The calcium sensor CBL2 and its interacting kinase CIPK6 are involved in plant sugar homeostasis via interacting with tonoplast sugar transporter TST2. *Plant Physiology*, 2020). So, we selected CBL1 and CBL2 subfamily genes for interaction analysis

between CIPK6s and CBL1/2s in *Gossypium*.

5. in Fig 4-d,4-e, both GhCIPK6D1 and GhCBL1A1 were detected in the nucleus and interacting between these two occurred in the nucleus, in addition to the PM. Were these signals real? The authors should explain its possible biological relevance.

Reply: Thank you for your comment. We re-validated the localization (Fig. 4e) and interaction between GhCBL1A1 and GhCIPK6D1 in cotton protoplasts (Fig. 4g), added the plasma membrane marker (CBLn:RFP) and nucleus marker (HY5:RFP), and cited the literature for nucleus marker in revised manuscript (Lin et al., B-BOX DOMAIN PROTEIN28 negatively regulates photomorphogenesis by repressing the activity of transcription factor HY5 and undergoes COP1-mediated degradation. *Plant Cell*, 2018).

The results showed that both GhCBL1A1 and GhCIPK6D1 were located to the plasma membrane and nucleus (Fig. 4d, e), while displayed high interaction signal on the plasma membrane other than on the nucleus (Fig. 4f, g).

6. In Fig 4-c, positive interactions between GhCBL1A1 and GhCIPK6D1, GhCIPK6D3 were detected. So what's the rationale to exclude the biological relevance of the GhCBL1A1 and GhCIPK6D3 complex?

Reply: Thank you for your comment. We confirmed that GhCBL2A1, rather than GhCBL1A1, affected the drought tolerance of GhCIPK6D3 through VIGS experiments (supplementary Fig. 16). The relevant description can be found in lines 331-336 of the article.

7. In Fig 8, tonoplast localization of GhCIPK6D3 and GhCBL2A1 should be demonstrated to support their function on vacuolar membranes.

Reply: The subcellular localization of GhCBL2A1 (formerly named GhCBL2) and GhCIPK6D3 (formerly named GhCIPK6), and the interaction between GhCBL2A1 and GhCIPK6D3 was reported in our previous study (Deng et al., The calcium sensor CBL2 and its interacting kinase CIPK6 are involved in plant sugar homeostasis via interacting with tonoplast sugar transporter TST2. *Plant Physiology*, 2020). It was reported that GhCBL2A1 was located to the vacuolar membrane, and GhCIPK6D3 was located to the plasma membrane and nucleus, BiFC and LUC experiments proved that they interact on the vacuole membrane (red arrows indication). These results were also cited in the revised manuscript.

[Redacted]

Deng et al., 2020 *Plant Physiol*

REVIEWER COMMENTS

Reviewer #1 (Remarks to the Author):

In the revised version of the manuscript, the authors have included some experiments that enhance the credibility of their assertions. Overall, this manuscript is well-written, coherent, and deserves to be published. However, there are a few minor points that could be addressed to further improve its quality.

1. Line 207-208, "Ri16 increased the expression of GhCIPK6A2/D2, but Ri19 showed the opposite trend (supplementary Fig.8)". Please explain the reasons for this difference.
2. Line 221-222, "The results indicated that GhCIPK6D1 had no impact on stomatal density..." Please rephrase the results. Additionally, in Supplementary Fig. 9, what does "n=5" specifically represent? The authors should provide explanations regarding the number of experimental repetitions, the quantity of leaves used per repetition, and the total number of stomata counted.
3. Line 264, Fig. 4f should be Fig. 4d-e; Line 266, Fig. 4g should be Fig. 4f-g.
4. Line 614, is it too few to have only 2-3 stomata? The reliability of results derived from limited data is subject to doubt.
5. In the legends of Figure 4, the unit for Bars should be μm instead of cm. Kindly review the entire document for analogous errors and rectify them.
6. In Supplementary Figure 1b, the significance of the numbers represented also requires elucidation in the figure legends.
7. The eighth page of the supplementary figures is unnecessary and should be removed.
8. Please also explain the meaning of "ns" in the figure legends.
9. In Supplementary Figure 18, there are typographical errors in GhCIBL1A1 and GhCIBL2A1 that need to be corrected.

Reviewer #2 (Remarks to the Author):

This is the manuscript I reviewed a while ago. After reading the author's response letter and the revised manuscript, I think they have done their best to address all of my previous concerns. I have no further questions.

Reviewer #3 (Remarks to the Author):

In this revised version, the authors have addressed several concerns raised by the reviewers. However, two crucial issues remain unresolved. To fully address these concerns is essential for the proposed working model.

1. the authors responded the issue on subcellular localization of GhCBL2-CIPK6 by stating that "It was reported that GhCBL2A1 was located to the vacuolar membrane, and GhCIPK6D3 was located to the plasma membrane and nucleus, BiFC and LUC experiments proved that they interact on the vacuole membrane (red arrows indication)."

I disagree with this response. LUC experiments do not tell the subcellular localization where two proteins interact. Resolution of BiFC assays was not sufficient to conclude that the interacting signals were from the PM. To make this claim, at least a tonoplast marker protein should be used in the BiFC assays, which is a very easy experiment to do.

2. the authors compared the expression of GhCIPK6D1, GhCIPK6D3, GhCBL1A1, and GhCBL2A1 between whole leaves and guard cells. They found that the transcript levels of these genes in guard cells are much lower than that in whole leaves (supplementary Fig. 18c) and thus concluded that the four genes are preferentially expressed in guard cells.

However, the conclusion should be opposite. If genes are preferentially expressed in guard cells, their transcript levels would much higher in guard cells than in whole leaves that contain a lot more other cells.

3. Strong signals of GhCIPK6D3-GFP were detected in the nucleus and OE24/35 have more stomata cells. Alternatively, so another hypothesis has emerged: CIPK6D3 can enter the nucleus to regulate stomata-differentiation-related gene expression by active/inactive TFs, such as FAMA, etc., and that more stomata cells in OE lines result in drought sensitivity.

Response to reviewers

We are grateful to you and the reviewers for the comments and suggestions on our manuscript. We have addressed each of the points raised, including carrying out additional work - our point-by-point responses are in red, below. We have uploaded a version of the manuscript that highlights all textual changes that address the reviewers' comments, and a clean version.

We hope the manuscript is suitable now for publication - we feel the suggestions have strongly improved the manuscript.

With best wishes

Prof. Xiyang Yang, on behalf of the authors.

Reviewer #1 (Remarks to the Author):

In the revised version of the manuscript, the authors have included some experiments that enhance the credibility of their assertions. Overall, this manuscript is well-written, coherent, and deserves to be published. However, there are a few minor points that could be addressed to further improve its quality.

1. Line 207-208, "Ri16 increased the expression of GhCIPK6A2/D2, but Ri19 showed the opposite trend (supplementary Fig.8)". Please explain the reasons for this difference.

Reply: Thank you for your comment. Gene expression is part of a complex regulatory network, and the expression of one gene might be regulated by multiple genes. In some cases, RNAi may affect these upstream genes, thus indirectly affecting the expression of target genes. According to our results, Ri16 and Ri19 showed drought sensitivity, but the expression of *GhCIPK6A2/D2* is different, so it could be concluded that the drought sensitivity of Ri16 and Ri19 is not related to the expression of *GhCIPK6A2/D2*.

2. Line 221-222, "The results indicated that GhCIPK6D1 had no impact on stomatal density..." Please rephrase the results. Additionally, in Supplementary Fig. 9, what does "n=5" specifically represent? The authors should provide explanations regarding the number of experimental repetitions, the quantity of leaves used per repetition, and the total number of stomata counted.

Reply: Thank you for your suggestions. We have rephrased the results in Line 221-222. In addition, in Supplementary Fig. 9, n=5 means five biological replicates that from five different plants of the same transgenic line, and selected the same leaf position, three fields were randomly selected for each biological replicate. We also described this in detail in the supplementary Fig. 9 legend and other relevant figures.

3. Line 264, Fig. 4f should be Fig. 4d-e; Line 266, Fig. 4g should be Fig. 4f-g.

Reply: Thank you for your suggestions. The results described the interaction between GhCBL1A1 and GhCIPK6D1 by BiFC in tobacco in line 264, so it matched Fig. 4f. And the results emphasized the strong interaction signal on the plasma membrane by BiFC in cotton protoplasts in line 266, so it matched Fig. 4g. We have a clearer description in the article for each figure.

4. Line 614, is it too few to have only 2-3 stomata? The reliability of results derived from limited data is subject to doubt.

Reply: Thank you for your comment. For K⁺ flux measurement, 2-3 stomata were randomly selected for measurement in each leaf, three biological replicates were measured and at least 23 leaves were selected by each biological replicate, so total 12-18 stomata were measured not 2-3.

5. In the legends of Figure 4, the unit for Bars should be μm instead of cm. Kindly review the entire document for analogous errors and rectify them.

Reply: Thank you for your suggestions. We made corrections and checked the entire documents carefully.

6. In Supplementary Figure 1b, the significance of the numbers represented also requires elucidation in the figure legends.

Reply: Thank you for your suggestions. The different numbers at the node of the evolutionary tree are the bootstrap value. They are used to assess the reliability of evolutionary tree branches. We have elucidated this in the legend.

7. The eighth page of the supplementary figures is unnecessary and should be removed.

Reply: Thank you for your suggestions. We have removed it.

8. Please also explain the meaning of "ns" in the figure legends.

Reply: Thank you for your comment. The "ns" means no significance, we have described it in the legend.

9. In Supplementary Figure 18, there are typographical errors in GhCIBL1A1 and GhCIBL2A1 that need to be corrected.

Reply: Thank you for your suggestions. We have corrected it.

Reviewer #2 (Remarks to the Author):

This is the manuscript I reviewed a while ago. After reading the author's response letter and the revised manuscript, I think they have done their best to address all of my previous concerns. I have no further questions.

Reviewer #3 (Remarks to the Author):

In this revised version, the authors have addressed several concerns raised by the reviewers. However, two crucial issues remain unresolved. To fully address these concerns is essential for the proposed working model.

1. the authors responded the issue on subcellular localization of GhCBL2-CIPK6 by stating that "It was reported that GhCBL2A1 was located to the vacuolar membrane, and GhCIPK6D3 was located to the plasma membrane and nucleus, BiFC and LUC experiments proved that they interact on the vacuole membrane (red arrows indication)."

I disagree with this response. LUC experiments do not tell the subcellular localization where two proteins interact. Resolution of BiFC assays was not sufficient to conclude that the interacting signals were from the PM. To make this claim, at least a tonoplast marker protein should be used in the BiFC assays, which is a very easy experiment to do.

Reply: Thank you for your comment. To prove that GhCBL2A1 interact with GhCIPK6D3 on the vacuole membrane, we performed experiments that localization of GhCBL2A1 and interaction with GhCIPK6D3 by BiFC in cotton protoplasts, and a tonoplast marker protein (RabG3b, a protein that has been reported to be localized in the vacuole membrane, Hawkins, T. et al. NET4 and RabG3 link actin to the tonoplast and facilitate cytoskeletal remodeling during stomatal immunity. *Nat. Commun.* 2023) be used (in supplementary Fig. 11). We have cited this literature in article. The results showed that GhCBL2A1 and GhCIPK6D3 were indeed interacting on the vacuole membrane. (The results as below)

Supplementary Fig. 11

2. the authors compared the expression of GhCIPK6D1, GhCIPK6D3, GhCBL1A1, and GhCBL2A1 between whole leaves and guard cells. They found that the transcript levels of these genes in guard cells are much lower than that in whole leaves (supplementary Fig. 18c) and thus concluded that the four genes are preferentially expressed in guard cells.

However, the conclusion should be opposite. If genes are preferentially expressed in guard cells, their transcript levels would much higher in guard cells than in whole leaves that contain a lot more other cells.

Reply: Thank you for your comment. We apologized for the misunderstanding caused by our statement. From our results, the four genes were highly expressed in leaf guard cells, and the expression levels of all four genes were higher than 50%. And we have rephrased the statement it in the article (Line 404-408).

3. Strong signals of GhCIPK6D3-GFP were detected in the nucleus and OE24/35 have more stomata cells. Alternatively, so another hypothesis has emerged: CIPK6D3 can enter the nucleus to regulate stomata-differentiation-related gene expression by active/inactive TFs, such as FAMA, etc., and that more stomata cells in OE lines result in drought sensitivity.

Reply: Thank you for your comment. There might be some other targets for GhCIPK6D3. However, from our results in Supplementary Fig. 9b, it was showed that OE24/35 had lower stomatal density (see below). So, we guessed that OE24/35 reduces stomatal opening and density, reduces water loss, and improves drought resistance, and we mainly did the molecular analysis on its function on stomatal movement in the current study.

Supplementary Fig. 9b

REVIEWERS' COMMENTS

Reviewer #1 (Remarks to the Author):

The authors have fully addressed my comments,I have no further comments.

Reviewer #3 (Remarks to the Author):

I have one minor concern for the revised version of the manu:

Overexpression of CIPK6D3(OE24/35) reduces not only stomatal opening but also density, both them are able to reduce water loss and improves drought resistance, however the authors claimed that stomatal movement is dominant reason for drought tolerance, which is arguable. I suggest the authors should recognize less stomatal density is also a non-negligible factor and write it in discussion.

Response to reviewers

We are grateful to you and the reviewers for the suggestions on our manuscript. We have addressed the point raised. We have uploaded a version of the manuscript that highlights all textual changes that address the reviewers' suggestions, and a clean version.

We hope the manuscript is suitable now for publication - we feel the suggestions have strongly improved the manuscript.

With best wishes

Prof. Xiyan Yang, on behalf of the authors.

Reviewer #1 (Remarks to the Author):

The authors have fully addressed my comments, I have no further comments.

Reviewer #3 (Remarks to the Author):

I have one minor concern for the revised version of the manu:

Overexpression of CIPK6D3(OE24/35) reduces not only stomatal opening but also density, both of them are able to reduce water loss and improve drought resistance, however the authors claimed that stomatal movement is the dominant reason for drought tolerance, which is arguable. I suggest the authors should recognize that less stomatal density is also a non-negligible factor and write it in the discussion.

Reply: Thank you for your suggestions. We have rediscussed these results in the discussion (Lines 450-454).